# MARché: Fast Masked Autoregressive Image Generation with Cache-Aware Attention

## Abstract

Masked autoregressive (MAR) models unify the strengths of masked and autoregressive generation by predicting tokens in a fixed order using bidirectional attention for image generation. Although effective, MAR models incur substantial computational overhead because they recompute attention and feed-forward representations for every token at each decoding step, even though the majority of tokens remain semantically unchanged across steps. We propose a training-free generation framework **MARché** to address this inefficiency through two key components: *cache-aware attention* and *selective KV refresh*. Cache-aware attention partitions tokens into active and cached sets, enabling separate computation paths that allow efficient reuse of previously computed key/value projections without compromising full-context modeling. However, a cached token cannot be used indefinitely without recomputation due to the changing contextual information over multiple steps. MARché recognizes this challenge and applies a technique called selective KV refresh. Selective KV refresh identifies contextually relevant tokens based on attention scores from newly generated tokens and updates only those tokens that require recomputation, while preserving image generation quality. MARché significantly reduces redundant computation in MAR without modifying the underlying architecture. Empirically, MARché achieves up to $1.7\times$ speedup with negligible impact on image quality, offering a scalable and broadly applicable solution for efficient masked transformer generation. The code is available at
`https://anonymous.4open.science/r/MARche-26F0`.

## 1 Introduction

Transformer models have achieved remarkable success in language generation (Vaswani et al., 2017; Brown et al., 2020; Raffel et al., 2020; Touvron et al., 2023; Radford et al., 2018; 2019; Ouyang et al., 2022), spurring their adoption in visual domains such as image synthesis. Autoregressive image generation models, in particular, generate pixels (Chen et al., 2020; Van den Oord et al., 2016; Reed et al., 2017) or tokens Parmar et al. (2018); Tian et al. (2024); Sun et al. (2024); Ramesh et al. (2021); Yu et al. (2022); Lee et al. (2022); Radford et al. (2021); Ding et al. (2021) sequentially by using causal attention to predict one token at a time conditioned on the past. Although effective, this strictly sequential process is inherently slow and, more importantly, ill-suited for capturing the two-dimensional spatial structure of images.

To overcome these limitations, masked generative models (Chang et al., 2022; 2023; Li et al., 2023) have emerged as a powerful alternative, characterized by parallel token generation and bidirectional attention. This leads to improved spatial coherence and significantly faster inference. Recently, masked autoregressive (MAR) models (Li et al., 2024) have demonstrated superior performance by combining the expressive power of autoregressive modeling with the efficiency of masked prediction. Unlike prior approaches that rely on vector quantization (Van Den Oord et al., 2017; Razavi et al., 2019; Esser et al., 2021), MAR operates in a continuous token space and employs a diffusion-based loss to model per-token distributions more effectively. Leveraging bidirectional attention within this framework, MAR achieves state-of-the-art image generation performance.

Despite its advantages, MAR, similar to masked generative approaches, still incurs substantial computational overhead. At each decoding step, the model recomputes attention and feed-forward representations for all tokens, even though only a small subset is selected for generation (also referred

to as the unmasked tokens). Empirically, we observe that a large number of key/value projections are stable across steps, showing minimal variation. This redundancy leads to inefficient computation that limits the scalability of MAR-based generation.

To address this inefficiency, we propose a novel, training-free generation approach **MARché** – MAR image generation with cache-aware attention. Our key insight is that token representations in MAR exhibit temporal locality: only a small subset of tokens (those currently being generated or contextually influenced by them) need to be updated, while the rest can safely reuse previously computed attention projections.

MARché consists of two core components: *cache-aware attention* and *selective KV refresh*. Cache-aware attention is an efficient mechanism that partitions tokens into *active* and *cached* sets, each following a separate computation path. The active set consists of tokens that are in need of computing their K and V values. The active set includes all the tokens that have been generated in the previous step and the tokens that are selected for generation in the current step. The cached set includes all the remaining tokens that have their K and V values computed in prior steps and will be reused. However, as we show in our empirical evaluations, reusing cached KV values indefinitely is detrimental to the generation quality. Hence, we exploit the structure of MAR to identify several contextually relevant tokens to be dynamically moved from cached set to the active set. We call this dynamic transition of token as the KV refresh component. We explicitly decouple the computation for active and cached tokens in both the attention and feed-forward layers, enabling efficient reuse of stable representations while preserving full bidirectional context.

Selective KV refresh complements cache-aware attention by efficiently identifying which tokens require recomputation. Inspired by previous works that use token-level correlation to manage KV caches (Xiao et al., 2023; Zhang et al., 2023; Ge et al., 2023; Adnan et al., 2024; Chen et al., 2024; Tang et al., 2024), we analyze attention scores from generating tokens and select only the most contextually relevant ones, defined by their high attention scores, as *refreshing tokens*. These tokens are then included in the active set. This lightweight mechanism preserves image generation quality while minimizing redundant computation.

Through extensive experiments, we demonstrate that MARché significantly accelerates MAR generation while maintaining high image fidelity. Compared to standard MAR, MARché achieves up to $1.7\times$ speedup while preserving generation quality. Importantly, MARché requires no architectural changes to the transformer and is fully compatible with existing MAR frameworks, making it broadly applicable in practice.

## 2 RELATED WORK

**Image generation with transformers.** Transformer models have been extended to image synthesis by leveraging discrete token representations (Van Den Oord et al., 2017; Razavi et al., 2019; Esser et al., 2021), which enable language-modeling techniques to be applied to visual data. Early approaches (Tian et al., 2024; Sun et al., 2024; Ramesh et al., 2021; Yu et al., 2022; Lee et al., 2022) adopt autoregressive decoding with causal attention, but struggle to model spatial dependencies. To address this, MaskGIT (Chang et al., 2022) proposes a masked generative model that predicts all tokens in parallel and refines them iteratively, offering improved spatial consistency and faster generation. Following this, recent masked generative models (Chang et al., 2023; Li et al., 2023; Weber et al., 2024; Lezama et al., 2022) further improve efficiency and flexibility. Among them, MAR (Li et al., 2024) stands out by unifying the strengths of autoregressive and masked modeling: it introduces bidirectional attention into a fixed-order autoregressive framework and eliminates the reliance on vector quantization through a diffusion-based loss. These optimizations allow MAR to achieve state-of-the-art image generation performance with both higher fidelity and faster inference, and form the baseline approach for this paper. Other methods, such as VAR (Tian et al., 2024), explore hierarchical prediction strategies, but follow a different direction.

**Efficient image generation.** Diffusion models have been widely studied from the perspective of efficiency, with strategies including accelerated sampling (Lu et al., 2022a;b; Liu et al., 2023), representation compression (Zhao et al., 2024b;a; Yuan et al., 2024), and intermediate result caching (Ma et al., 2024; Wimbauer et al., 2024; Agarwal et al., 2024). In contrast, efficiency in transformer-based generative models has received less attention despite their high-quality outputs. For autoregressive

transformers, speculative decoding (Jang et al., 2024; Teng et al., 2024) enables partial parallelism by sampling ahead and verifying predictions, but it is constrained to left-to-right generation and thus orthogonal to our masked setting.

Within masked generative models, several recent approaches have explored efficiency from different perspectives. ENAT (Ni et al., 2024) accelerates decoding by modeling spatial and temporal dependencies between tokens but requires training a dedicated attention module, while our approach is training-free. LazyMAR (Yan et al., 2025) speeds up MAR by detecting redundant token updates through hidden feature similarity. Instead of leveraging the KV cache, LazyMAR reuses hidden features. As it adaptively selects tokens based on feature similarity, it departs from MAR's predefined generation order, potentially altering the intended generation trajectory, while our approach preserves the order. MaGNeTS (Goyal et al., 2025) improves generation efficiency by dynamically scaling model size during decoding and caching KV pairs, but its caching is tightly integrated with the model's progressive resizing strategy.

KV caching is proposed to improve the efficiency of auto-regressive models (Brown et al., 2020; Radford et al., 2019; Xiao et al., 2023; Zhang et al., 2023; Ge et al., 2023; Adnan et al., 2024; Chen et al., 2024; Tang et al., 2024; Shen et al., 2021). However, the MAR paradigm was not designed to exploit KV caching since the $K$ and $V$ values of a token are repeatedly updated on each step of MAR. MARché makes the key observation that KV projections of many tokens remain stable across steps, and hence KV caching can in fact be adapted to the MAR strategy to improve efficiency. However, the cached KV values have to be occasionally recomputed when the attention scores of cached tokens change. MARché exploits these observations to improve the MAR paradigm without requiring any additional training or architectural modification. Crucially, we decouple the attention mechanism into recomputation and reuse phases, enabling selective KV refresh based on token-level relevance. By leveraging the structural properties and generation semantics of MAR, MARché achieves both computational efficiency and architectural generality in a lightweight, training-free manner.

# 3 PRELIMINARY: MASKED AUTOREGRESSIVE MODEL

MAR models generate images by iteratively predicting a subset of masked tokens based on previously known ones, following a randomly permuted order. This approach generalizes traditional autoregressive decoding by allowing multiple predictions per step while maintaining the autoregressive nature of next-token (or next-set-of-tokens) prediction.

**Formulation.** Let $x = (x_1, x_2, \ldots, x_N)$ denote a sequence of image tokens to be generated. At the start of inference, all tokens are unknown and initialized as masked, i.e., $x_i^{(0)} = \texttt{[MASK]}$ for all $i$, and the initial set of unknown positions is $M^{(0)} = \{1, 2, \ldots, N\}$. A random permutation $\pi$ of the index set $\{1, 2, \ldots, N\}$ defines the generation order, which decides the subset of positions $U^{(t)} \subset M^{(t)}$ to be predicted at each generation step, where $M^{(t)}$ is the current set of masked token positions. $U^{(t)}$ follows this fully randomized order, rather than being chosen adaptively.

The generation proceeds in two stages: encoding and decoding. The encoder $f_{\text{enc}}$ processes only the known tokens, i.e., those at positions $i \notin M^{(t)}$, embedding each with its positional encoding $\text{PE}_i$. This yields contextual features for the visible part of the sequence:

$$h_{\text{enc}}^{(t)} = f_{\text{enc}}(\{x_i^{(t)} + \text{PE}_i \mid i \notin M^{(t)}\}). \tag{1}$$

To form the decoder input, we combine $h_{\text{enc}}^{(t)}$ with embeddings $h_{\text{mask}}^{(t)}$ for the masked tokens (i.e., positions in $M^{(t)}$), aligned with their respective positional encodings. The decoder $f_{\text{dec}}$ attends to the entire sequence and outputs contextual vectors:

$$z^{(t)} = f_{\text{dec}}(\text{concat}(h_{\text{enc}}^{(t)}, h_{\text{mask}}^{(t)}), \text{PE}). \tag{2}$$

For each selected position $i \in U^{(t)}$, a token value is sampled from the conditional distribution modeled by the diffusion sampler, conditioned on the decoder output $z_i^{(t)}$:

$$x_i^{(t+1)} \sim p_\theta(x_i \mid z_i^{(t)}; \tau), \tag{3}$$

where $\tau$ is a temperature parameter that controls the diversity of the generated samples. The predicted values replace the corresponding masked tokens, and the set of unknown positions is updated:

$$M^{(t+1)} = M^{(t)} \setminus U^{(t)}. \tag{4}$$

This iterative generation process continues, progressively reducing the number of masked positions, until the entire token sequence is completed, i.e., $M^{(T)} = \emptyset$.

**Motivation.** Although MAR models follow an autoregressive decoding paradigm, they use bidirectional attention at each decoding step, which allows *all* tokens, including both generated and yet to be generated ones, to update their contextual representations. This design enables global context exchange, but also implies that the features of *all* tokens are recomputed at every step.

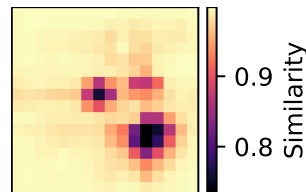

However, since only a small subset of tokens is newly generated at each step, the contextual information of the majority of tokens varies minimally between steps. To quantify this redundancy, we analyze the cosine similarity of key projections between consecutive steps. As shown in Figure 1, a large number of tokens exhibit consistently high similarity, often exceeding 0.95, indicating that their internal representations remain nearly unchanged. We observe a similar trend for value projections, reinforcing this observation.

Figure 1: **Cosine similarity of key projections** between step 4 and 5 (in layer 2) during a single image generation. The x- and y-axes represent token indices.

These findings suggest that fully recomputing attention representations for all tokens at every step is inefficient. By identifying and reusing stable token representations, we can significantly reduce computational cost.

## 4 MARCHÉ: CACHE-AWARE ATTENTION WITH SELECTIVE KV REFRESH

As observed in Section 3, many tokens in MAR generation exhibit stable KV projections throughout steps. This temporal redundancy suggests an opportunity to reduce the computation by caching and reusing KV projections. However, not all tokens are safe to cache, as some exhibit noticeable changes in their contextual representations across steps.

To address this challenge, we introduce two key ideas. (1) *cache-aware attention*, which separates tokens into *active* and *cached* sets, assigning them to distinct computation paths during attention and feed-forward layers (Section 4.1). (2) *Selective KV cache refresh*, augments active tokens by adding *refreshing tokens*. These tokens are not being generated but are contextually influenced by generation targets. To efficiently identify them, we propose a lightweight attention-guided selection strategy, described in Section 4.2.

### 4.1 CACHE-AWARE ATTENTION MECHANISM

**Token categorization: active vs. cached.** At each generation step, we categorize tokens into two groups: *active* tokens and *cached* tokens. Active tokens are those whose contextual representations must be updated; cached tokens are considered to have very limited changes to their KV projections and can safely reuse their cached key/value (KV) projections.

The active set includes three types of tokens: (1) *Generating tokens*, which are masked tokens selected for prediction at the current step; (2) *Caching tokens*, which are newly generated tokens from the previous step that have not yet been incorporated into the cache; and (3) *Refreshing tokens*, which are not currently being generated but are contextually influenced by the new predictions and thus require updated projections. Note that these definitions are used consistently throughout the paper and play a central role in the ablation analysis in Section 5.3.

While generating and caching tokens can be identified deterministically, refreshing tokens are selected based on their correlation with generating tokens, as detailed in Section 4.2. All remaining tokens are treated as cached. This categorization forms the foundation of our efficient decoding approach: instead of recomputing all token representations, cache-aware attention assigns computation only to tokens that truly require it.

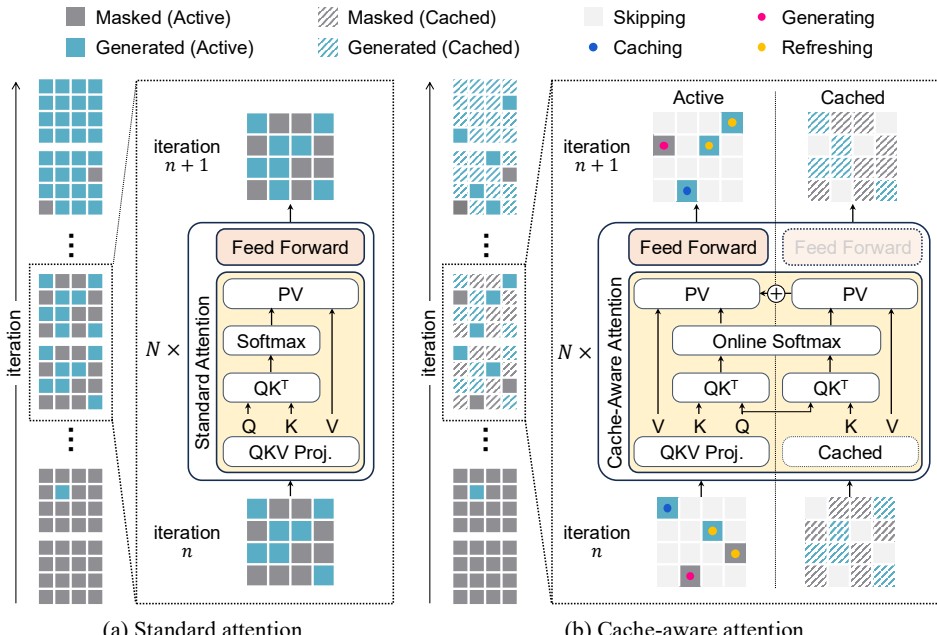

Figure 2: **Comparison of standard attention and cache-aware attention in MAR generation.** **(a)** In standard attention, all tokens (masked or generated) are processed at every step, leading to redundant computation. **(b)** In cache-aware attention, only *active tokens* (generating, refreshing, or caching) are recomputed, while *cached tokens* reuse previously computed key/value projections.

**Cache-aware attention design.** As illustrated in Figure 2, cache-aware attention achieves efficiency by splitting the attention operation into two computation paths—one for active tokens and one for cached tokens. Active tokens project fresh queries, keys, and values, and compute attention over the union of both active and cached tokens. Cached tokens, by contrast, do not generate new projections and are not used as queries, but their stored key/value projections contribute as context during attention. This separation allows us to reuse representations where possible, without sacrificing the bidirectional nature of the model.

For each active token, attention is computed in two parts. In the first, queries attend to freshly computed keys and values from the active set ($K_A$, $V_A$); in the second, the same queries attend to cached keys and values from previous steps ($K_C$, $V_C$). To avoid data movement overhead from concatenation, we compute these two attention scores independently and merge the results using an *safe online softmax* formulation (Dao et al., 2022; Dao, 2023; Shah et al., 2024). With $\alpha^{(A)}, \alpha^{(C)}$, and $\ell_i$ from the online softmax, the output vector is computed as a sum of contributions from active and cached tokens, avoiding the need for key/value concatenation. This ensures exact equivalence to standard attention while enabling better memory locality and kernel fusion.

After attention, only active tokens are forwarded through the feed-forward network, while cached tokens are skipped entirely in this stage. Finally, the key/value projections of active tokens are written back to the cache. These design choices of query-side separation, feed-forward skipping, and selective cache updates enable significant runtime savings without modifying the underlying transformer architecture. The full algorithm is summarized in Algorithm 1. We provide a mathematical formulation and proof of equivalence in Appendix B, and report a thorough analysis of speedups in Appendix C and D.

## 4.2 SELECTIVE KV CACHE REFRESH

**Criteria for selecting refreshing tokens.** As discussed in Figure 1, although most tokens exhibit stable KV representations across steps, a subset of tokens plays a crucial role in maintaining image generation quality—particularly those influenced by newly generated content. Naturally, newly generated tokens must be recomputed. However, tokens that are influenced by the newly unmasked tokens may also need to be refreshed, as their importance can shift dynamically during generation.

---

**Algorithm 1** Cache-aware Attention

---

**Require:** Token embeddings $x^{(t)}$, cache $\mathcal{C}^{(t)}$, generation mask $M^{(t)}$
1: Identify generating tokens $G^{(t)} \subset M^{(t)}$
2: Identify caching tokens $N^{(t)}$ from previous step
3: Identify refreshing tokens $R^{(t)}$ based on correlation with $G^{(t)}$
4: Set active set $A^{(t)} = G^{(t)} \cup R^{(t)} \cup N^{(t)}$, cached set $C^{(t)} = \{1, \ldots, N\} \setminus A^{(t)}$
5: Compute $Q, K, V$ for $i \in A^{(t)}$; retrieve cached $K, V$ for $i \in C^{(t)}$
6: **for all** $i \in A^{(t)}$ **do**
7:     Compute attention logits over $A^{(t)}$ and $C^{(t)}$, apply safe online softmax:
$$\alpha^{(A)}, \alpha^{(C)}, \ell_i = \texttt{SafeOnlineSoftmax}(q_i, K_A, K_C)$$
8:     Compute output vector: $z_i = \frac{\alpha^{(A)} V_A + \alpha^{(C)} V_C}{\ell_i}$
9:     Apply feed-forward: $x_i^{(t+1)} = \text{FFN}(z_i^{(t)})$
10: **end for**
11: Update cache: $\mathcal{C}^{(t+1)}[i] = (K_i, V_i)$ for $i \in A^{(t)}$

---

To identify such tokens, we draw inspiration from autoregressive language models (Xiao et al., 2023; Zhang et al., 2023; Ge et al., 2023; Adnan et al., 2024; Chen et al., 2024; Tang et al., 2024), where attention scores are commonly used to determine which cached key/value entries can be safely discarded or ignored during inference. In contrast, we repurpose attention scores in our setting to do the opposite: to identify which tokens should be actively refreshed. By selecting tokens that receive high attention from newly generated tokens, we can target the refresh process toward contextually important tokens. This strategy enables us to maintain image generation quality with minimal recomputation. We validate its effectiveness in Section 5.3.

**Design of selective KV cache refresh.** Building on our empirical findings, we design a selective KV cache refresh mechanism that dynamically identifies which tokens require recomputation based on their contextual relevance to the generating tokens. As illustrated in Figure 3, we leverage attention scores computed in the second decoder layer (Layer 2). Note that the choice of decoder layer for this selection is configurable, allowing for a trade-off between speedup and image quality, which we discuss further in Section 5.3.

At each generation step, Layers 1 and 2 perform standard full attention, as the active tokens are not yet finalized. During the Layer 2 attention computation, attention scores from generating tokens to all other tokens are computed and aggregated across all attention heads to obtain a global relevance score for each token. We then select the top-$K$ tokens with the highest relevance scores as the refreshing tokens. These refreshing tokens, together with the newly generated tokens and the previously cached tokens, form the active token set for subsequent layers. In these layers, cache-aware attention is applied: representations are recomputed only for active tokens, while the KV projections of non-active (cached) tokens are reused without reprocessing. This targeted refresh strategy enables the model to preserve contextual consistency while minimizing computational overhead.

### 4.3 IMPLEMENTATION DETAILS

**Updating active tokens.** At each generation step, the active token set always includes tokens that are selected for generation in the current step (generating) and tokens selected in the immediately prior step (caching). The attention scores of the first decode layer are then used to select the top-K tokens (refreshing), such that the active set fits a fixed batch size of 64. For example, if there are 5 generating tokens and 9 caching tokens selected, we select the remaining 50 tokens as the refresh tokens based on the attention scores. The number of refreshing tokens is thus adjusted accordingly to fit within this budget. An ablation study on this value is provided in Appendix E.

**Refreshing entire KV Cache.** We refresh the entire KV cache in three specific cases to ensure correctness and stability. First, during the initial step, since no KV projections have been generated yet, we apply full attention across all layers. Second, in every step, the first and second decoder layers perform full attention as the active tokens are not yet finalized. Lastly, to mitigate potential value drift

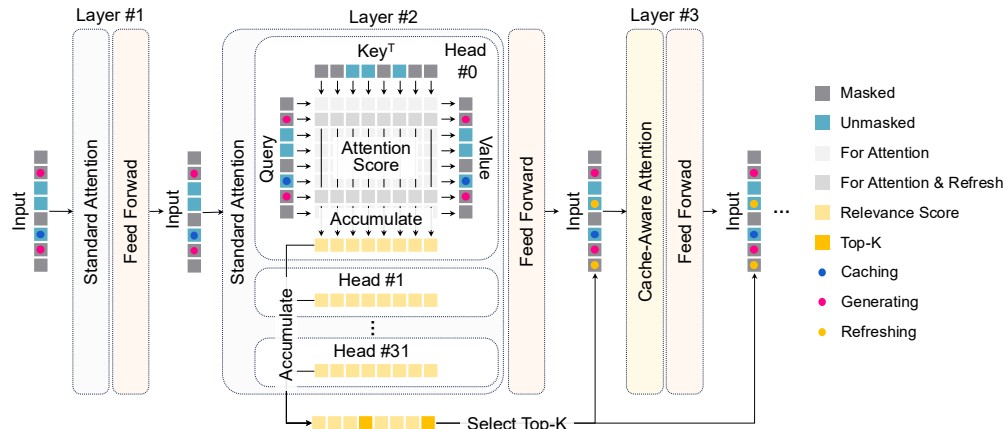

Figure 3: **KV cache refresh via attention score.** Layers 1 and 2 perform standard full attention. In Layer 2, attention scores are computed from generating tokens to all others, and aggregated across attention heads to produce global relevance scores. The top-$K$ most contextually relevant tokens are selected based on these scores and marked as refreshing tokens. From Layer 3 onward, cache-aware attention is applied.

caused by repeated caching, we periodically refresh the entire KV cache every three steps, a process we refer to as periodic full refresh. Ablation results analyzing the impact of full refresh frequency and the choice of refresh layers are provided in Appendix F and Appendix G, respectively.

## 5 EVALUATION

We evaluate MARché along three key dimensions: (1) image quality, to ensure that generation fidelity is preserved; (2) inference latency, to assess computational efficiency; and (3) design choices, to analyze how different components of our method impact performance.

### 5.1 EXPERIMENTAL SETUP

We evaluate our proposed MARché framework on the ImageNet dataset at $256 \times 256$ resolution in the class-conditional setting. We adopt three model scales from the original MAR implementation (Li et al., 2024): MAR-B (Base), MAR-L (Large), and MAR-H (Huge). For each model, we construct a corresponding MARché variant (MARché-B, MARché-L, MARché-H) by applying our cache-aware inference strategy without modifying the model architecture or training procedure.

To assess image quality, we report Fréchet Inception Distance (FID) (Heusel et al., 2017) and Inception Score (IS) (Salimans et al., 2016). Computational efficiency is measured using the latency metric (in seconds per image). Speedup is calculated as the ratio of latency between the original MAR and its corresponding MARché variant at each model scale.

We compare MARché against several strong baselines: MaskGIT (Chang et al., 2022), a masked generative transformer; DiT (Peebles & Xie, 2023), a diffusion transformer; LlamaGen (Sun et al., 2024), an autoregressive model; and MAR (Li et al., 2024). All experiments are conducted on a single NVIDIA H100 GPU. We use a batch size of 128 for main evaluations and 256 for ablation studies. We follow the default MAR inference schedule of 64 decoding steps.

### 5.2 MAIN RESULTS

Table 1 presents a comprehensive comparison of our MARché models against recent high-performing image generation baselines, including MaskGIT (Chang et al., 2022), DiT (Peebles & Xie, 2023), and LlamaGen (Sun et al., 2024), as well as the original MAR models across three different scales.

Across all model sizes, MARché achieves substantial reductions in inference latency while maintaining competitive generation quality. For instance, MARché-B reduces latency from 0.104s to 0.064s

Table 1: **Comparison of MARché and baseline models on ImageNet 256×256.** MARché achieves up to 1.72× speedup over the original MAR models while maintaining competitive image quality.

| Method | Latency (s/im) ↓ | FID ↓ | IS ↑ | Param | Speedup ↑ |
|---|---|---|---|---|---|
| MaskGIT (Chang et al., 2022) | 0.440 | 6.18 | 182.1 | 227M | - |
| DiT-XL/2 (Peebles & Xie, 2023) | 0.787 | 2.27 | 278.2 | 675M | - |
| LlamaGen-XXL (Sun et al., 2024) | 0.897 | 3.09 | 253.6 | 1.4B | - |
| LlamaGen-3B (Sun et al., 2024) | 1.011 | 3.05 | 222.3 | 3.1B | - |
| MAR-B (Li et al., 2024) | 0.104 | 2.35 | 281.1 | 208M | 1.00 |
| MARché-B | 0.064 | 2.56 | 270.3 | 208M | 1.57 |
| MAR-L (Li et al., 2024) | 0.193 | 1.84 | 296.3 | 479M | 1.00 |
| MARché-L | 0.115 | 2.16 | 278.6 | 479M | 1.68 |
| MAR-H (Li et al., 2024) | 0.336 | 1.62 | 298.6 | 943M | 1.00 |
| MARché-H | 0.188 | 2.02 | 281.4 | 943M | 1.79 |

per image, achieving a 1.57× speedup over MAR-B, with negligible change in FID (from 2.35 to 2.56) and Inception Score (from 281.1 to 270.3). Similar trends are observed for the larger models: MARché-L and MARché-H show 1.68× and 1.72× speedups, respectively, while maintaining image quality within acceptable margins. Appendix H presents qualitative comparisons showing that MARché-H produces images that are visually indistinguishable from those of MAR-H. The FID and IS scores of MARché variants remain close to those of the original MAR models, and are consistently better than those of MaskGIT and LlamaGen. While DiT achieves slightly better FID than MARché-B, it does so at the cost of significantly higher latency.

Overall, MARché offers a compelling trade-off between generation speed and quality, outperforming baseline masked and autoregressive models in both inference efficiency and scalability. Importantly, these gains are achieved without modifying the original architecture or retraining, demonstrating the practical applicability of our approach. We further provide a Pareto curve in Appendix H, which visualizes the quality–efficiency trade-off across different model scales.

Table 2: **Effect of token selection strategies for constructing the active set.** The results illustrate generation performance with respect to the inclusion of *generating tokens* and *caching tokens*, as defined in Section 4.1.

| Strategy | FID ↓ |
|---|---|
| MARché | **2.56** |
| MARché w/o *caching tokens* | 2.70 |
| MARché w/o *generating tokens* | 504.82 |
| Random selection | 564.61 |

Table 3: **Effect of refreshing token selection strategy.** Our MARché approach, which selects tokens receiving high attention from generating tokens, outperforms both low-attention and random selection strategies, demonstrating the effectiveness of relevance-based refreshing.

| Strategy | FID ↓ |
|---|---|
| MARché | **2.56** |
| MARché w/ low attention scores | 3.80 |
| MARché w/ random selection | 3.01 |

## 5.3 ABLATION STUDY

**Ablation on token types in active set construction.** We validate the design of our active token selection strategy in MARché by comparing four different approaches in Table 2. The compared strategies are: (1) the full MARché method; (2) MARché without *caching tokens*; (3) MARché without *generating tokens*; (4) random selection of tokens for recomputation.

As shown in Table 2, the full MARché configuration achieves the best performance with an FID of 2.56. Including caching tokens results in some performance improvement, and more importantly, that inclusion costs no additional computational burden. In contrast, excluding generating tokens leads to a substantial degradation in quality, confirming their essential role in preserving semantic consistency during decoding. Finally, using randomly selected tokens for recomputation yields the worst performance, highlighting the importance of guided, attention-based selection.

These results demonstrate that generating tokens are indispensable for accurate generation, while incorporating caching tokens offers additional benefits with minimal computational cost.

**Correlations of refreshing tokens and attention scores.** We evaluate the impact of different strategies for selecting *refreshing tokens*. As shown in Table 3, we compare three approaches.

The first strategy follows MARché's approach, selecting refreshing tokens with high attention scores to generating tokens. The second selects tokens with low attention scores to generating tokens, and the third randomly selects a subset of tokens for refreshing, ignoring contextual relevance.

As the results indicate, selecting tokens based on low attention scores leads to worse performance than random selection, with a clearly higher FID. This suggests that refreshing tokens should be chosen based on their strong correlation with generating tokens in order to maintain high image generation quality.

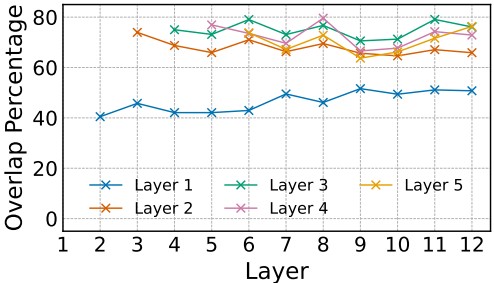 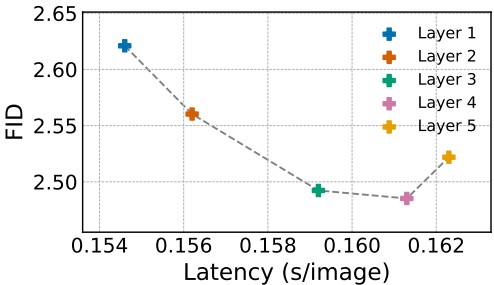

Figure 4: **Top-$K$ attention overlap across decoder layers.** Deeper layers (3–5) show higher consistency overall, while Layer 1 exhibits much lower overlap (49.3%).

Figure 5: **Latency–FID trade-off for different refreshing token selection layers.** Deeper layers improve FID with higher latency, while earlier layers decode faster but degrade quality.

**Choosing the decoder layer for refreshing token selection.** We analyze the impact of selecting different decoder layers for computing attention scores used in refreshing token selection. This decision involves a trade-off between computational efficiency and generation quality, as it affects both which tokens are refreshed and where full attention is computed.

Figure 4 shows the overlap ratio between layer 1-5 and all subsequent layers. As shown in the figure, deeper layers tend to produce refreshing token selections that align more closely with other layers. Specifically, Layer 3 achieves the highest average overlap of 74.4% with the rest, while Layer 1 shows lower alignment at 49.3%. Layer 2 exhibits a moderate level of agreement at 66.8%, suggesting it captures context reasonably well while still remaining computationally efficient.

This trade-off is also reflected in generation performance. Figure 5 shows that using Layer 1 results in faster decoding (0.154 s/image) but yields a relatively high FID of 2.62. On the other hand, selecting a deeper layer like Layer 4 improves FID to 2.49 but with increased latency (0.1613 s/image). Layer 2 provides a balanced compromise, attaining an FID of 2.56 with moderate latency (0.1593 s/image).

In our experiments, we adopt Layer 2 as a default, as it provides a favorable trade-off between speed and quality while maintaining reasonable alignment with other layers' token selection.

## 6 CONCLUSION

We presented MARché, a cache-aware decoding framework for masked autoregressive image generation that significantly reduces inference latency while maintaining high image quality. By selectively refreshing only contextually relevant tokens based on attention scores, MARché avoids redundant computation and achieves up to $1.7\times$ speedup across model scales without modifying the original architecture. Our approach highlights the potential of combining autoregressive modeling with efficient token reuse strategies in high-resolution image generation. While our method focuses on masked autoregressive models, the core ideas behind cache-aware attention and selective refresh may extend to other generative settings, including multi-modal generation.

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

# A  THE USE OF LARGE LANGUAGE MODELS

We use large language models (LLMs) to aid or polish the writings. Specifically, LLMs were employed to improve readability, polish grammar, and refine phrasing. All conceptual contributions, experimental design, implementation, and analysis were carried out by the authors.

# B  EQUIVALENCE OF CACHE-AWARE ATTENTION TO STANDARD ATTENTION

We show that our cache-aware attention mechanism is mathematically equivalent to standard full attention, assuming no approximation in cache usage or token selection. This guarantees that our method maintains the expressivity of full bidirectional attention despite operating in a more efficient, partitioned manner.

Note that our implementation employs a customized kernel that integrates both safe online softmax and FlashAttention. In cache-aware attention, the key-value tokens are partitioned into two disjoint subsets: the active token set and the cached token set. To correctly merge the attention contributions from these subsets, MARché uses safe online softmax. Within each subset's computation, MARché further leverages FlashAttention, which efficiently partitions queries (Q), keys (K), and values (V) into blocks and schedules these blocks across GPU multiprocessors via Triton. In the following, we focus on the cache-aware attention process with safe online softmax, since FlashAttention is applied identically in both standard and cache-aware attention.

## B.1  STANDARD ATTENTION FORMULATION

Let $q_i \in \mathbb{R}^d$ be a query corresponding to token $i$, and $K = [k_1, \ldots, k_N]^\top \in \mathbb{R}^{N \times d}$, $V = [v_1, \ldots, v_N]^\top \in \mathbb{R}^{N \times d}$ be the full sets of key and value vectors. The standard scaled dot-product attention computes:

$$\text{Attn}(q_i, K, V) = \sum_{j=1}^{N} \frac{\exp\left(\frac{q_i \cdot k_j}{\sqrt{d}}\right)}{\sum_{\ell=1}^{N} \exp\left(\frac{q_i \cdot k_\ell}{\sqrt{d}}\right)} v_j. \tag{5}$$

## B.2  CACHE-AWARE ATTENTION FORMULATION

In cache-aware attention, we partition the key-value set into two disjoint subsets:

- $A$: the active token set, with freshly computed $K_A, V_A$,
- $C$: the cached token set, with previously computed $K_C, V_C$,

such that $A \cup C = \{1, \ldots, N\}$ and $A \cap C = \emptyset$.

Rather than computing attention over the full $K, V$ simultaneously, we compute the attention output incrementally using the online softmax trick as follows:

$$s^{(A)} = \frac{q_i K_A^\top}{\sqrt{d}}, \quad s^{(C)} = \frac{q_i K_C^\top}{\sqrt{d}}, \tag{6}$$

$$m_i = \max(\max s^{(A)}, \max s^{(C)}), \tag{7}$$

$$\alpha^{(A)} = \exp(s^{(A)} - m_i), \quad \alpha^{(C)} = \exp(s^{(C)} - m_i), \tag{8}$$

$$\ell_i = \sum \alpha^{(A)} + \sum \alpha^{(C)}, \tag{9}$$

$$z_i = \frac{\alpha^{(A)} V_A + \alpha^{(C)} V_C}{\ell_i}. \tag{10}$$

## B.3  EQUIVALENCE PROOF

Observe that standard attention can also be rewritten in terms of a shared max $m_i$:

$$\text{Attn}(q_i) = \frac{\sum_{j=1}^{N} \exp\left(\frac{q_i \cdot k_j}{\sqrt{d}} - m_i\right) v_j}{\sum_{j=1}^{N} \exp\left(\frac{q_i \cdot k_j}{\sqrt{d}} - m_i\right)}. \tag{11}$$

Since $A$ and $C$ partition $\{1, \ldots, N\}$, the sums over $j \in A \cup C$ in cache-aware attention cover exactly the same elements as the full attention formulation. Therefore,

$$z_i = \frac{\sum_{j \in A \cup C} \exp\left(\frac{q_i \cdot k_j}{\sqrt{d}} - m_i\right) v_j}{\sum_{j \in A \cup C} \exp\left(\frac{q_i \cdot k_j}{\sqrt{d}} - m_i\right)} = \text{Attn}(q_i, K, V).$$

### B.4 Conclusion

Thus, the cache-aware attention formulation yields exactly the same output as standard attention, up to floating point precision. The separation into active and cached components, combined with the use of an online softmax normalization, offers significant computational advantages without sacrificing model fidelity.

## C Runtime Efficiency of Cache-Aware Attention Implementation

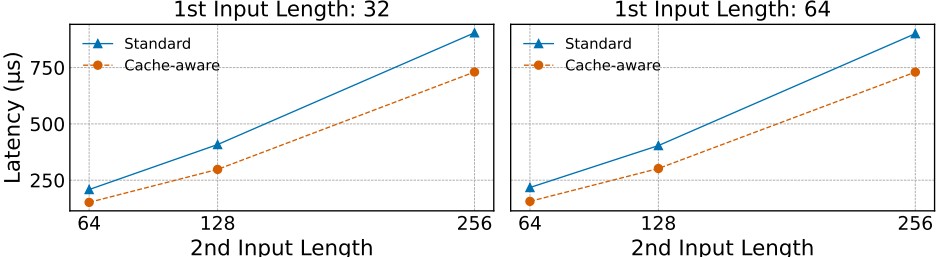

Figure 6: **Runtime comparison of standard attention vs. cache-aware attention implementation.** Both methods process the same total number of attention operations over two input segments of different lengths. *Standard attention* concatenates the inputs and processes them jointly, while *cache-aware attention* handles them separately.

In the cache-aware attention operation, the inputs include the query vector $q_i$, active keys and values $K_A$ and $V_A$, and cached keys and values $K_C$ and $V_C$. A vanilla implementation leverages a standard attention kernel by first concatenating $K_A$ with $K_C$, and $V_A$ with $V_C$, to form unified key and value projections. These are then passed to the standard attention kernel for computation. However, we demonstrate that this approach is suboptimal. Assuming all inputs $q_i$, $K_A$, $V_A$, $K_C$, and $V_C$ are preloaded and available, we compare the latency of this baseline with our optimized cache-aware attention kernel.

Figure 6 reports kernel-level latency across varying input sizes. Despite involving the same total number of attention operations, our cache-aware attention kernel consistently achieves lower latency than the vanilla concatenation-based implementation. For instance, when the two input lengths are 32 and 256, cache-aware attention reduces latency from $904.2\mu s$ to $730.3\mu s$, yielding a 19.2% improvement. Across all tested configurations, we observe latency reductions ranging from 16.2% to 27.3%, with larger input asymmetries showing greater benefit.

These improvements arise primarily from leveraging a contiguous memory layout. Cache-aware attention addresses memory inefficiencies by separating active and cached token sets into distinct computation paths. After identifying generating, caching, and refreshing tokens, only the active tokens are gathered into a contiguous set, while the remaining tokens form the cached set. Processing the two sets in parallel removes the need for concatenation and avoids costly non-contiguous memory

accesses during attention computation. By employing an efficient memory layout, cache-aware attention achieves substantial latency gains and overall speedups.

## D  SPEEDUP CONTRIBUTIONS OF CACHE-AWARE ATTENTION

Cache-aware attention provides latency gains through efficient memory management and the reuse of cached keys and values. In addition, it enables skipping part of the feed-forward layer (FFN) computation, since cached tokens do not require regeneration. We refer to this as FFN skipping. Therefore, we identify two components contributing to the overall speedup: (1) cache-aware attention itself and (2) FFN skipping. To disentangle their contributions, we implement four different methods, each utilizing either cache-aware attention, FFN skipping, or both, and evaluate MAR with each implementation.

Table 4: **Latency and speedup comparison across different implementations**, isolating the effects of cache-aware attention and FFN skipping.

| Method | Implementation | Latency (s) | Speedup |
|---|---|---|---|
| MAR | Standard attention + full FFN | 0.104 | 1× |
| MAR w/ selected FFN | Standard attention + FFN skipping | 0.092 | 1.12× |
| MAR w/ cache-aware attention | Cache-aware attention + full FFN | 0.067 | 1.55× |
| MARché | Cache-aware attention + FFN skipping | 0.064 | 1.63× |

Table 4 reports the latency and speedup of each implementation. The results indicate that cache-aware attention contributes primarily to the speedup of MARché, while FFN skipping also provides a smaller yet meaningful improvement. Based on these results, we confirm that cache-aware attention significantly reduces the computation path for the cached token set and enables more efficient memory management, making it the key factor behind the observed speedup.

## E  EFFECTIVENESS OF THE NUMBER OF REFRESHING TOKENS

We analyze how the number of refreshing tokens affects the trade-off between generation quality and latency. In MARché, the number of refreshing tokens is determined by the total number of active tokens minus the generating and caching tokens. By varying the number of active tokens, we can control how many tokens are refreshed at each step.

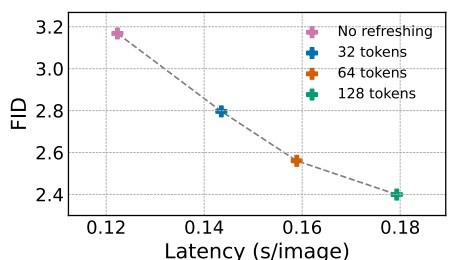

Figure 7: **Trade-off between latency and FID as a function of the number of refreshing tokens.** Using more refreshing tokens improves FID but increases latency.

As shown in Figure 7, increasing the number of refreshing tokens improves image quality, lowering FID, but also increases latency due to additional computation. Among the tested settings, using 64 refreshing tokens provides a good balance, achieving low FID with moderate latency.

We also evaluate a configuration with no refreshing tokens, which yields the fastest inference (0.12 s/image), but results in a notable drop in image quality, with FID rising to 3.17. This highlights the necessity of refreshing contextually relevant tokens for maintaining generation fidelity.

## F  EFFECT OF REFRESH FREQUENCY ACROSS STEPS

By default, MARché performs a full refresh of the KV cache every 3 decoding steps. We investigate how the frequency of these periodic refreshes affects generation quality and inference speed. Specifically, we vary the refresh period (i.e., how often full refresh is applied during decoding) and evaluate the trade-off between FID and latency.

As shown in Figure 8, shorter refresh cycles (e.g., every 2 steps) yield better image quality with lower FID scores, but incur higher latency due to more frequent computation. In contrast, longer cycles (e.g., every 6–7 steps) reduce latency but significantly degrade generation quality. Notably, a 3-step refresh period offers a favorable trade-off: it achieves competitive FID while being much faster than 2-step refresh. Based on this, we adopt a 3-step refresh as our default configuration.

These results highlight the importance of carefully tuning refresh frequency. Overly frequent refresh leads to redundant computation, while infrequent refresh compromises semantic consistency in generation.

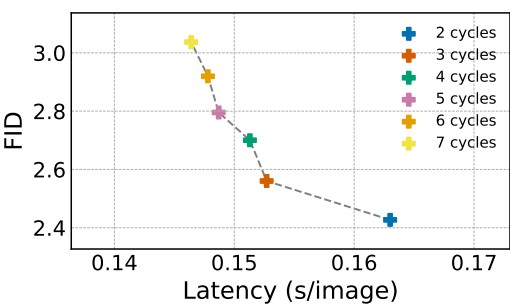

Figure 8: **Trade-off between refresh frequency and generation performance.** Shorter refresh cycles (e.g., every 2–3 steps) improve FID at the cost of increased latency, while longer cycles reduce latency but degrade image quality. A 3-step refresh strikes a good balance, achieving low FID with faster decoding.

## G    EFFECT OF REFRESH LOCATION ACROSS LAYERS

We conduct an ablation study to examine how the placement of full refresh across decoder layers affects generation quality and inference speed in MARché. Table 5 compares four strategies in which full refresh is applied to different subsets of layers over 12 decoding layers. Note that Layer 1 always performs full refresh across all strategies, as it is used to select refreshing tokens.

Table 5: **Ablation on full refresh placement across layers.** We compare four layer-wise full refresh strategies over 12 decoding layers. Refreshing only Layer 1 yields the fastest inference, while refreshing Layers 1–6 slightly improves quality (lowest FID and highest IS). Refreshing deeper or non-consecutive layers degrades both quality and speed.

| Refresh Strategy | Latency (s/im) ↓ | FID ↓ | IS ↑ |
|---|---|---|---|
| Full at layer 1 only | **0.155** | 2.62 | 266.3 |
| Full for layers 1–6 | 0.164 | **2.49** | **275.1** |
| Full for layers 7–12 | 0.177 | 25.35 | 98.50 |
| Full on even layers | 0.185 | 4.65 | 212.3 |

Applying full refresh only at Layer 1 yields the fastest inference (0.158 s/im) while maintaining strong image quality (FID 2.56, IS 270.3). Refreshing the early layers (Layers 1–6) slightly improves quality (FID 2.49, IS 275.1) at a small latency cost (0.160 s/im). In contrast, refreshing the deeper layers (Layers 7–12) leads to significant degradation in FID (25.35) and IS (98.5), along with increased latency. A similar trend is observed when refreshing only the even-numbered layers.

These results suggest that, when full refresh is applied, it is most effective to perform it in the early layers and in consecutive order, rather than scattered or at later layers.

## H    QUALITATIVE COMPARISON ON GENERATED IMAGES

We conduct a qualitative comparison between standard MAR and our proposed MARché, using the MAR-H and MARché-H, respectively. As shown in Figure 9, despite MARché-H achieving a significant 1.72× speedup over MAR-H, the visual quality of the generated images remains comparable. The outputs from MARché-H are virtually indistinguishable from those of MAR, demonstrating that our cache-aware generation approach preserves image fidelity while substantially improving efficiency. This highlights MARché's practical value in scenarios requiring both high-quality generation and fast inference.

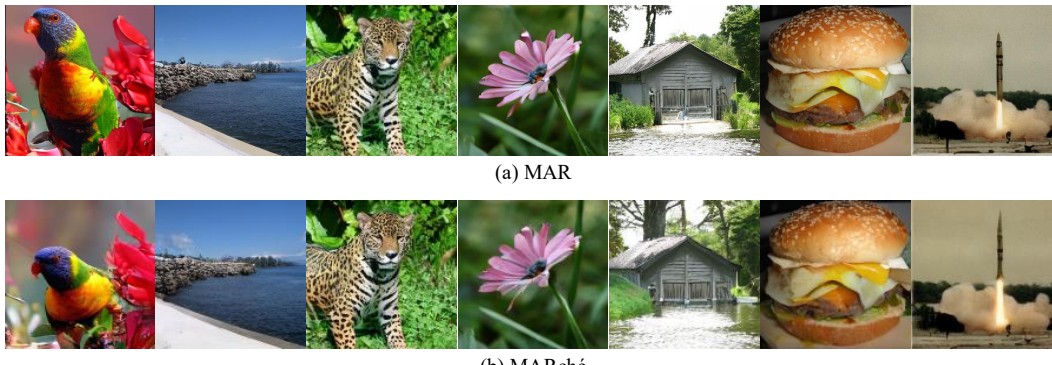

(a) MAR

(b) MARché

Figure 9: **Qualitative comparison of generated images from MAR-H (top) and MARché-H (bottom).** MARché-H achieves similar visual quality while running 1.7× faster.

# I    PARETO CURVE OF LAZYMAR VS. MARCHÉ

To further illustrate the trade-off between generation efficiency and image quality, we visualize the Pareto frontier of LazyMAR and MARché variants across different model scales (B, L, and H) in Figure 10. The x-axis denotes inference latency (seconds per image), while the y-axis reports FID scores on ImageNet 256×256.

As shown, MARché consistently shifts the Pareto frontier toward the lower-left region of the curve, achieving faster inference and improved FID compared to LazyMAR at all scales. For instance, MARché-B reduces latency from 0.074s to 0.064s (1.16× faster) while also improving FID (from 5.32 to 2.56). A similar trend holds for larger models: MARché-L achieves a 1.15× speedup with lower FID, and MARché-H provides a 1.17× speedup while improving FID from 4.00 to 2.02,

Overall, the Pareto visualization demonstrates that MARché achieves a uniformly better quality–efficiency balance than LazyMAR across all tested scales. MARché requires no retraining or architectural modifications, and its cache-aware attention with selective KV refresh yields consistently superior efficiency and image quality, highlighting its effectiveness as a general and scalable approach for masked autoregressive generation.

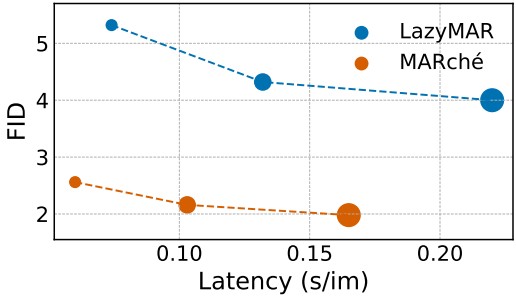

Figure 10: **Pareto frontier of LazyMAR and MARché variants on ImageNet 256×256 across different model scales.** MARché consistently achieves faster inference with improved FID, shifting the efficiency–quality trade-off toward a more favorable region compared to LazyMAR.

# J    STABILITY OF KEY/VALUE PROJECTIONS ACROSS LAYERS AND STEPS

To further understand the temporal redundancy of token representations in MARché, we visualize the cosine similarity of key and value projections between decoding steps across different decoding layers. Figures 11 and 12 show pairwise similarities for selected steps and layers.

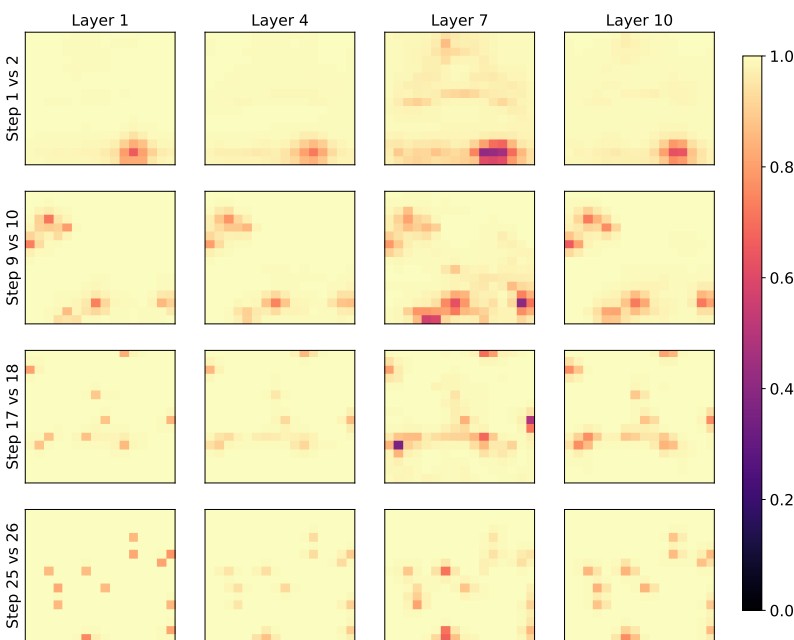

Figure 11: **Cosine similarity of key projections across decoding steps and layers.** Most tokens maintain high similarity across steps, especially in lower layers, indicating that their key projections change very little during decoding.

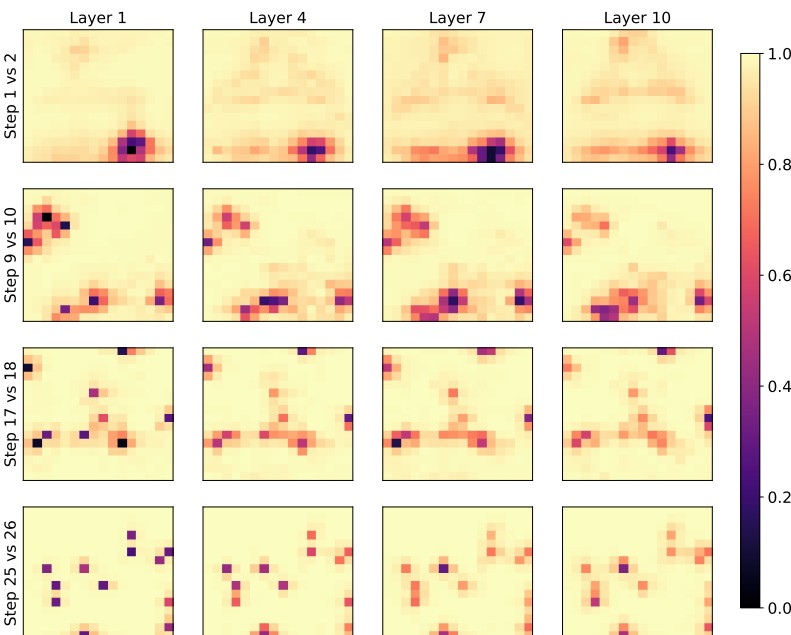

Figure 12: **Cosine similarity of value projections across decoding steps and layers.** Value projections are slightly more dynamic than key projections, particularly in deeper layers, but remain stable for the majority of tokens.

**Key projections.** As shown in Figure 11, the key projections remain highly stable throughout generation, especially in shallow layers (e.g., Layer 1 and Layer 4). Even in deeper layers, most

tokens exhibit similarities above 0.9 between steps, suggesting minimal variation in their contextual embeddings. This supports our motivation that many tokens do not require recomputation at each step.

**Value projections.** Figure 12 reveals slightly more variability in the value projections, particularly in deeper layers and at later steps. Nonetheless, the majority of tokens still exhibit high similarity across steps, reinforcing the notion that value projections are also largely stable during decoding.

These findings confirm that both key and value representations show strong temporal locality. This justifies the design of cache-aware attention and selective KV refresh, as only a small subset of tokens truly require recomputation during generation.

## K  LAYER-WISE CONSISTENCY OF REFRESHING TOKEN SELECTION OVER STEP

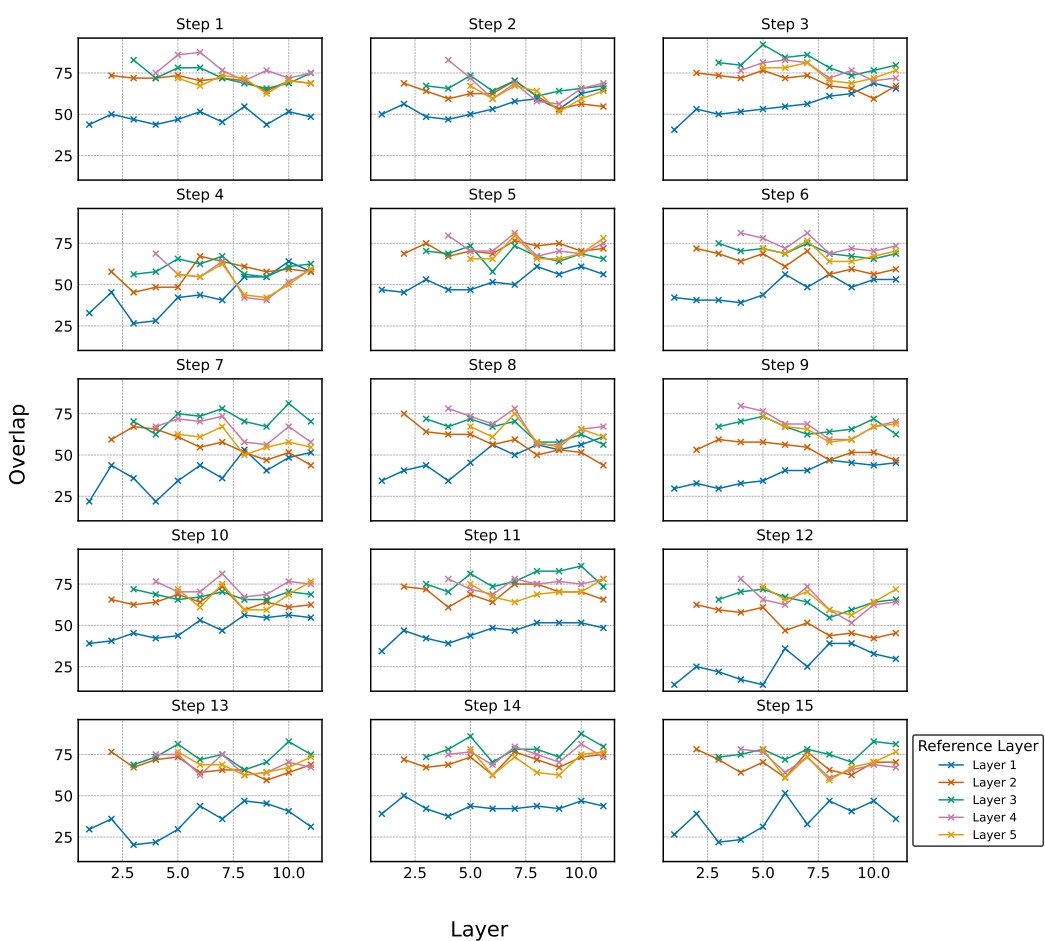

Figure 13: **Layer-wise top-$K$ attention overlap across decoding steps.** Deeper layers show higher and more consistent alignment.

To further analyze the stability of refreshing token selections across layers, we visualize the overlap of top-K attended tokens across decoder layers at each generation step (Steps 1–15), as shown in Figure 13. This complements the aggregated results in Figure 4 by revealing temporal dynamics of layer-wise agreement.

Each subplot in Figure 13 corresponds to a generation step and plots the overlap percentage of different decoder layers with a given reference layer. Across all steps, we observe that deeper layers

(Layers 3–5) consistently exhibit higher mutual alignment compared to shallower ones. Notably, Layer 1 shows significantly lower agreement with other layers, often falling below 50% overlap. In contrast, Layer 3 maintains consistently strong overlap with other deep layers, reaffirming its role as a stable and contextually rich candidate for refreshing token selection.

Layer 2 emerges as a strong middle ground: it achieves higher consistency than Layer 1 while avoiding the computational burden of deeper layers like Layer 4 or 5. Its overlap trends remain reasonably stable across steps, frequently aligning above 60% with deeper layers, which suggests that it captures contextual signals effectively without incurring excessive computation. This observation supports our choice of Layer 2 as the default configuration, offering a practical trade-off between stability and efficiency for refreshing token selection.

## L  EFFECT OF BATCH SIZE ON LATENCY

We evaluate the effect of batch size on latency by measuring MAR-B and MARché-B with batch sizes ranging from 32 to 512. As shown in Table 6, the latency of both methods increases with larger batch sizes. Across all settings, MARché achieves consistent speedups over MAR, ranging from $1.27\times$ to $1.61\times$. These results demonstrate that MARché consistently provides substantial latency reduction compared to MAR.

Table 6: **Latency comparison of MAR and MARché across different batch sizes.** MARché consistently achieves faster inference, with speedups ranging from $1.27\times$ to $1.61\times$.

| Batch Size | MAR (s/im) | MARché (s/im) | Speedup ↑ |
|:---:|:---:|:---:|:---:|
| 32 | 0.0283 | 0.0222 | $1.27\times$ |
| 64 | 0.0535 | 0.0368 | $1.45\times$ |
| 128 | 0.1038 | 0.0664 | $1.56\times$ |
| 256 | 0.2063 | 0.1570 | $1.31\times$ |
| 512 | 0.4133 | 0.2560 | $1.61\times$ |