# OpenReview forum: "MARché: Fast Masked Autoregressive Image Generation with Cache-Aware Attention"
_ICLR.cc/2026/Conference — Submitted to ICLR 2026_

### Official Review · Reviewer_L52V · 2025-10-26

**Soundness:** 3
**Presentation:** 3
**Contribution:** 3
**Rating:** 6
**Confidence:** 4

**Summary:**

This manuscript introduces "MARché," a training-free inference acceleration framework specifically designed to optimize the image generation process of Masked Autoregressive (MAR) models. MAR models exhibit significant computational redundancy during inference: attention and feed-forward network (FFN) computations are recalculated for all tokens at each step, even though most tokens remain unchanged. MARché addresses this by leveraging KV Cache through two key components:

1. Cache-aware Attention: Partitions tokens into "Active" and "Cached" sets, designing distinct computation paths (Active tokens recompute QKV and pass through FFN; Cached tokens only provide KV and skip FFN) while maintaining mathematical equivalence.

2. Selective KV Refresh: Recognizes that caches cannot be used indefinitely. This mechanism analyzes attention scores from the 2nd decoder layer to identify and select the Top-K cached tokens most relevant to the currently generating tokens as "Refreshing Tokens," adding them to the active set for forced updates. It also incorporates periodic full refresh (default: every 3 steps) to prevent error accumulation.

Without modifying the MAR model structure or requiring retraining, MARché significantly reduces inference computation. Experiments on the ImageNet 256x256 dataset show that MARché provides up to 1.7x (approaching 1.8x for MAR-H) inference speedup for MAR models, with minimal impact on generation quality (FID/IS).

**Strengths:**

1. **Inference Acceleration**: The core contribution is practical and impactful, achieving speedups of 1.57x to nearly 1.8x for MAR models (Table 1).

2. **Training-Free**: As a pure inference-time optimization, MARché is easy to adopt and applicable to existing pretrained MAR models without costly retraining.

3. **Preservation of Generation Quality**: The method maintains high fidelity, with only slight increases in FID while significantly reducing latency (Table 1, Fig 9).

4. **Clever Mechanism Design**: The combination of cache-aware attention (with efficient kernel implementation) and attention-guided selective KV refresh is well-motivated and technically sound.

5. **Clear Exposition and Thorough Experimentation**: The paper effectively explains the redundancy problem in MAR models and clearly details the MARché design. It provides comprehensive experimental validation, including extensive ablation studies, strongly supporting the methodology and the role of each component.

**Weaknesses:**

1. **Dependence on MAR Architecture**: MARché is specifically optimized for MAR models (with fixed generation order and bidirectional attention). It's unclear if this optimization can be extended to other types of masked generative models (like MaskGIT with potentially different generation strategies) or models without a fixed order. Its generality might be limited by the MAR architecture itself.

2. **Hyperparameter Sensitivity**: The effectiveness of selective KV refresh seems dependent on several key hyperparameters, such as the layer chosen for selecting refresh tokens, the number K of refresh tokens (dynamic strategy vs. fixed value), and the frequency of periodic full refresh. While ablations are provided, choosing these hyperparameters might still require careful tuning for specific applications.

3. **Insufficient Comparison with Similar Methods**: The paper mentions another MAR acceleration work, LazyMAR (Yan et al., 2025), and notes differences (reusing hidden states vs. KV cache, changing vs. preserving generation order), but lacks a direct performance comparison (speedup vs. FID). This makes it difficult for readers to assess the relative advantages of MARché among similar approaches.

**Questions:**

**Generality of the Refresh Token Selection Layer:**

Question: The paper chooses the 2nd layer's attention scores to determine refresh tokens, showing it's a good speed-quality balance (Fig 5). Is this choice optimal across different model scales (MAR-L, MAR-H) and potentially other datasets or tasks? For instance, might deeper models benefit from scores derived from a deeper layer (e.g., Layer 3, whose Top-K selection aligns more closely with even deeper layers, see Fig 4) to maintain quality? Discussion or experiments on generality are suggested.

**Comparison of Dynamic K vs. Fixed K Refresh Strategies:**

Question: The current strategy dynamically adjusts the number of refresh tokens (K) to fill a batch size of 64. How does performance (speed vs. FID) change if a strategy with a fixed K (e.g., always refreshing the Top-50 tokens) is adopted? Does the dynamic K strategy lead to significant fluctuations in computation per step? Which strategy is preferable for practical deployment?

**Potential for Adaptive Periodic Full Refresh Frequency:**

Question: The current approach uses a fixed 3-step cycle for full refresh. Considering that token changes might differ across generation stages (e.g., early stages generating outlines vs. late stages filling details), could an adaptive full refresh strategy (e.g., dynamically adjusting frequency based on token change rate or generation stage) further optimize performance?

**Applicability of MARché to Other Bidirectional Attention Models:**

Question: The core idea of MARché (identifying and caching less-changing KVs) doesn't seem strictly limited to MAR models with a fixed order. To what extent do the authors believe this method could be applied to other iterative generation models employing bidirectional attention (e.g., certain types of image editing models, non-autoregressive iterative refinement for language models, or MaskGIT models with non-fixed generation orders)? What would be the main challenges?

---

> ### Author Response · Authors · 2025-11-21
> **Response to Reviewer L52V (1/3)**
>
> ## Generality of MARché  (W1, Q4)
>
> Thank you for raising this important question regarding the generality of our method.
>
> We would first like to clarify that existing masked generative models such as MaskGIT are not designed with KV caching in mind.
> In MaskGIT, all previously generated tokens are recomputed at every iteration because their representations are not stable across cycles.
> This instability fundamentally prevents the use of KV caching: once a token is generated, its hidden representation does not remain consistent enough to be safely reused.
>
> In contrast, although MAR was also not originally designed for caching, its fixed generation order and monotonically expanding set of generated tokens provide structural opportunities for KV reuse.
> Once a token is unmasked in MAR, it remains unmasked for the rest of the decoding process, which creates a stable generation trajectory.
> However, because MAR uses bidirectional attention, the representations (and thus KV values) of previously generated tokens can still fluctuate as new tokens are introduced.
> Our analysis (Figures 11 and 12) shows that these fluctuations are sparse, most tokens remain highly stable across steps.
>
> This observation is precisely what enables MARché.
> We propose a principled mechanism to (1) identify which tokens require recomputation and which can be safely cached, and (2) perform efficient attention by separating tokens into cached and active sets.
> This makes KV caching feasible even under full bidirectional attention, which typically prevents such reuse.
>
> Given that MAR currently represents the state-of-the-art masked autoregressive framework for image generation, we believe that enabling efficient KV caching within this setting is a meaningful contribution.
> Our method demonstrates that bidirectional masked generative models, which have traditionally been viewed as incompatible with caching, can, in fact, benefit significantly from it when analyzed appropriately.
>
> Finally, while we agree that MARché is directly applicable only to MAR-like architectures (i.e., models with a fixed generation order and stable generated tokens), we believe that other masked generative models could be redesigned to support KV caching.
> If future architectures explicitly encourage stability of generated tokens and adopt a predictable generation schedule, the core ideas of MARché could be extended to them as well.
> We view this as an exciting direction for future work.
>
>
> ## Comparison with Similar Method (W3)
>
> We appreciate the reviewer’s suggestion and have conducted a comprehensive comparison against LazyMAR under the exact same experimental settings as our MARché evaluation.
> As shown in the table below, MARché consistently outperforms LazyMAR across all model scales.
> LazyMAR achieves a 1.41×–1.53× speedup, whereas MARché achieves a larger 1.57×–1.72× speedup while also producing better image quality (lower FID and higher IS).
> This demonstrates that MARché offers a more favorable quality–latency trade-off among existing training-free MAR acceleration methods.
>
> Since the LazyMAR source code was released on October 24, after our submission deadline, it was not feasible to include a comparison at submission time.
> We will include these results in the revised paper, along with a Pareto curve to clearly highlight the trade-off between LazyMAR and MARché.
>
> |Model Size|Model|Latency|FID ↓|IS ↑|Speedup|
> |--|--|--|--|--|--|
> |Base|LazyMAR-B|0.074|5.32|235.1|1.41×|
> ||MARché-B|0.064|2.56|270.3|1.57×|
> |Large|LazyMAR-L|0.132|4.32|246.6|1.46×|
> ||MARché-L|0.115|2.16|278.6|1.68×|
> |Huge|LazyMAR-H|0.220|4.00|251.9|1.53×|
> ||MARché-H|0.195|2.02|281.4|1.72×|

---

> ### Author Response · Authors · 2025-11-21
> **Response to Reviewer L52V (2/3)**
>
> ## Hyperparameter Design Rationale and Stability of the Dynamic-K Strategy (W2, Q2)
>
> While several hyperparameters in MARché are empirically chosen, their values are not arbitrary.
> They are motivated by concrete observations about the dynamics of MAR decoding.
> For example, the active set size of 64 is determined based on the generation schedule of MAR: each step unmasks only 6–7 new tokens, so the combined number of generating and caching tokens remains below 14 across all steps (below Table).
> This leaves more than 50 slots available for refreshing tokens, allowing a substantial refresh budget without sacrificing the benefits of KV reuse.
>
> Our analysis (Figures 11 and 12) further shows that only a small subset of tokens undergoes large changes in contextual (KV) representation across steps.
> The sparsity of such fluctuations is consistent across model sizes, suggesting that refreshing roughly 20\% of tokens per step is sufficient to capture the important updates while maintaining fast inference.
>
> Regarding the reviewer’s question on dynamic-K vs. fixed-K strategies, our dynamic approach, filling the active set up to the fixed budget of 64, was specifically chosen to maintain stable computation per step.
> Because the numbers of generating and caching tokens slightly vary across steps, dynamically adjusting the refresh token count ensures that the total active-set size remains constant, making the computational cost predictable and the implementation simple.
> In contrast, a fixed-K strategy would use a constant number of refreshing tokens, which means the size of the active set would fluctuate over time, resulting in step-dependent computation and less predictable latency.
> For practical deployment, we therefore view the dynamic-K strategy, which keeps the active-set size and per-step cost constant, as more attractive.
>
> | Step | B | H | Step | B | H | Step | B | H | Step | B | H |
> |------|---|---|------|---|---|------|---|---|------|---|---|
> | 1 | 1 | 1 | 17 | 2 | 2 | 33 | 5 | 5 | 49 | 6 | 6 |
> | 2 | 1 | 1 | 18 | 3 | 3 | 34 | 5 | 5 | 50 | 5 | 5 |
> | 3 | 1 | 1 | 19 | 3 | 3 | 35 | 4 | 4 | 51 | 6 | 6 |
> | 4 | 1 | 1 | 20 | 3 | 3 | 36 | 5 | 5 | 52 | 6 | 6 |
> | 5 | 1 | 1 | 21 | 3 | 3 | 37 | 5 | 5 | 53 | 6 | 6 |
> | 6 | 1 | 1 | 22 | 3 | 3 | 38 | 5 | 5 | 54 | 6 | 6 |
> | 7 | 1 | 1 | 23 | 3 | 3 | 39 | 5 | 5 | 55 | 6 | 6 |
> | 8 | 1 | 1 | 24 | 4 | 4 | 40 | 5 | 5 | 56 | 6 | 6 |
> | 9 | 1 | 1 | 25 | 3 | 3 | 41 | 5 | 5 | 57 | 7 | 7 |
> | 10 | 1 | 1 | 26 | 4 | 4 | 42 | 6 | 6 | 58 | 6 | 6 |
> | 11 | 1 | 1 | 27 | 4 | 4 | 43 | 5 | 5 | 59 | 6 | 6 |
> | 12 | 1 | 1 | 28 | 4 | 4 | 44 | 5 | 5 | 60 | 6 | 6 |
> | 13 | 1 | 1 | 29 | 4 | 4 | 45 | 6 | 6 | 61 | 6 | 6 |
> | 14 | 2 | 2 | 30 | 4 | 4 | 46 | 5 | 5 | 62 | 7 | 7 |
> | 15 | 3 | 3 | 31 | 4 | 4 | 47 | 6 | 6 | 63 | 6 | 6 |
> | 16 | 2 | 2 | 32 | 4 | 4 | 48 | 6 | 6 | 64 | 6 | 6 |
>
> ## Generality in Selecting the Refresh-Token Layer (Q1)
>
> Based on the active-set structure described above, the choice of which layer’s attention scores to use for refresh-token selection is treated as a hyperparameter.
> As the reviewer noted, this choice reflects a balance between inference speed and image quality.
> For MAR-B, we select Layer 2 as our default because it provides the best trade-off between latency and accuracy.
>
> To evaluate the generality of this choice, we additionally examined deeper layers in MAR-L, as summarized in the table below.
> We observe that using deeper layers indeed improves FID and IS monotonically, consistent with the reviewer’s suggestion and with the alignment trends shown in Fig. 4.
> However, deeper layers also incur higher latency, resulting in a less favorable overall speed–quality balance.
> For this reason, Layer 2 also remains the preferred choice for MAR-L.
>
> While we did not repeat this sweep for MAR-H or other datasets due to computational cost, we expect a similar trend to hold:
> deeper layers provide marginally better quality, whereas shallower ones offer better speed, and Layer 2 typically sits near the practical optimum.
> We will include a discussion on this generality consideration in the revised version.
>
> | Layer | Latency (s) | FID    | IS        |
> |-------|-------------|--------|-----------|
> | 1     | 0.1090      | 2.36 | 272.1  |
> | 2     | 0.1148      | 2.16 | 278.6  |
> | 3     | 0.1193      | 2.11 | 282.9  |
> | 4     | 0.1226      | 2.07 | 283.7  |
> | 5     | 0.1267      | 2.06 | 284.9  |

---

> ### Author Response · Authors · 2025-11-21
> **Response to Reviewer L52V (3/3)**
>
> ## Full-Refresh Schedule (Q3)
>
> Thank you for raising this insightful point regarding the possibility of an adaptive full-refresh schedule.
>
> As the reviewer suggests, different stages of MAR decoding may produce different patterns of token changes—for example, early stages generating coarse structures and later stages refining details. Our analysis in Figures 11 and 12 indeed shows that the spatial distribution of high-variation tokens varies between early and late stages.
>
> However, two observations indicate that these stage-dependent changes do not significantly affect MARché’s efficiency or image quality:
>
> First, although the locations of high-variation tokens shift, the number of such tokens remains consistently small across layers and decoding steps. Thus, the proportion requiring recomputation stays well within the capacity of our active set.
>
> Second, MARché always refreshes the top-ranked unstable tokens at every step. This ensures that whenever a region undergoes a sudden semantic transition, the relevant tokens naturally receive higher refresh priority. In this sense, the selective-refresh mechanism already adapts dynamically to token-change patterns without needing an explicit stage-dependent rule.
>
> Based on these observations, we believe the fixed 3-step full-refresh cycle is sufficient to maintain stable quality while keeping the design simple.
>
> Nevertheless, we agree that an adaptive full-refresh strategy, one that adjusts frequency based on token-change statistics or generation stage, could potentially yield a better accuracy–latency trade-off. Investigating such adaptive schemes is an interesting direction for future work, and we appreciate the reviewer for highlighting it.

---

> > ### Comment · Reviewer_L52V · 2025-11-23
> >
> > Thanks for the explanation. The authors have answered my concern, but I don't see the revised version that includes the ablation with LazyMAR and 'a Pareto curve' mentioned above.

---

> > > ### Author Response · Authors · 2025-11-25
> > >
> > > Thank you for pointing this out. We have updated the revised version accordingly. Specifically, we added the Pareto curve comparing LazyMAR and MARché in the appendix (Appendix I) and highlighted the changes in the manuscript. The curve is constructed based on the results provided in the comparison table included in our response to Comment W3.

---

### Official Review · Reviewer_54c7 · 2025-10-29

**Soundness:** 2
**Presentation:** 3
**Contribution:** 2
**Rating:** 4
**Confidence:** 4

**Summary:**

This paper presents MARché a training-free acceleration framework for Masked Autoregressive (MAR) image generation models. The method addresses computational redundancy in MAR models through two core components: (1) cache-aware attention that partitions tokens into active and cached sets with independent computational paths, and (2) selective KV refresh that identifies context-relevant tokens for recomputation based on attention scores. Experimental results demonstrate that MARch achieves 1.7× inference speedup on ImageNet 256×256 with negligible impact on image quality.

**Strengths:**

1. The KV caching mechanism is innovatively adapted from autoregressive language models for masked generation tasks, effectively solving the cache staleness issue via selective refresh. The approach is elegantly designed, with its implementation details clearly articulated.
2. The method is architecture-agnostic and training-free, making it easily integrable into existing MAR frameworks.

**Weaknesses:**

1. The method incorporates multiple empirical hyperparameters (e.g., fixed active set size of 64, full refresh every 3 steps, use of Layer 2 attention scores). Although ablation studies demonstrate their effectiveness, deeper theoretical analysis or principled design guidance is lacking. For instance, is a fixed size of 64 always optimal across varying image complexities or generation steps? It is suggested to discuss the robustness of these hyperparameters in different scenarios or explore an adaptive mechanism for determining the active set size.
2. Missing analysis of memory overhead: While KV caching accelerates computation, it inevitably increases memory usage. The paper does not quantify or discuss the additional memory overhead introduced by MARché.
3. Experiments are conducted only on ImageNet 256×256, lacking validation on higher-resolution datasets such as 512×512 or 1024×1024.
4. Insufficient comparison with recent works: The paper mentions LazyMAR in the related work section but does not include it as a baseline in the experiments. A direct performance (speed/quality) comparison with LazyMAR is needed.Additionally, while the proposed method achieves a 1.7× speedup, the IS metric decreases significantly, which is not observed with LazyMAR. Please explain the reason for this discrepancy.
5. The discussion of limitations in the conclusion is relatively brief. For example, during stages where the generated content changes abruptly (e.g., transitioning from background to foreground objects), the stability of KV projections may decrease. Could this affect the efficiency and quality of MARché?

**Questions:**

1. Although ablation studies demonstrate their effectiveness, deeper theoretical analysis or principled design guidance is lacking. For instance, is a fixed size of 64 always optimal across varying image complexities or generation steps?
2. The discussion of limitations in the conclusion is relatively brief. For example, during stages where the generated content changes abruptly (e.g., transitioning from background to foreground objects), the stability of KV projections may decrease. Could this affect the efficiency and quality of MARché?

---

> ### Author Response · Authors · 2025-11-21
> **Response to Reviewer 54c7 (1/2)**
>
> ## Hyperparameter Design Rationale and Robustness (W1, Q1)
>
> Thank you for raising this important point regarding the hyperparameter choices and their underlying design principles.
>
> While several of our hyperparameters are empirically selected, they are not arbitrary.
> Their values are motivated by observations about the dynamics of MAR decoding, and we designed MARché to operate robustly under these characteristics.
>
> First, the active set size of 64 is informed by the structure of MAR’s generation schedule.
> Each step unmasks only a small number of tokens, typically no more than 6–7 tokens, as shown in below Table.
> Thus, the combined number of generating and caching tokens per step remains below 14 for both the base and huge models.
> The remaining active capacity can therefore be dedicated to refreshing tokens.
>
> Second, as shown in Figures 11 and 12, only a small subset of tokens exhibits large contextual (KV) changes across sequential steps.
> This sparsity motivates assigning a sufficient refresh budget.
> An active set of 64 enables us to refresh more than 50 tokens per step, covering roughly 20\% of all tokens, which is large enough to capture those with significant variation while still allowing substantial KV reuse for the rest.
>
> These considerations ensure that MARché can safely reuse KV caches under bidirectional attention, enabling efficient decoding without compromising the model’s representational flexibility.
> Within this framework, we calibrated additional hyperparameters, such as performing a full refresh every 3 steps and using Layer 2 attention scores for identifying refreshing tokens, by balancing speedup and image quality.
> As Figures 4, 5, and 8 demonstrate, these choices represent strong trade-off points.
>
> We appreciate the reviewer for highlighting a direction that can further strengthen our work.
> In the revised version, we will add a detailed discussion on the robustness of these hyperparameters.
>
> | Step | B | H | Step | B | H | Step | B | H | Step | B | H |
> |------|---|---|------|---|---|------|---|---|------|---|---|
> | 1 | 1 | 1 | 17 | 2 | 2 | 33 | 5 | 5 | 49 | 6 | 6 |
> | 2 | 1 | 1 | 18 | 3 | 3 | 34 | 5 | 5 | 50 | 5 | 5 |
> | 3 | 1 | 1 | 19 | 3 | 3 | 35 | 4 | 4 | 51 | 6 | 6 |
> | 4 | 1 | 1 | 20 | 3 | 3 | 36 | 5 | 5 | 52 | 6 | 6 |
> | 5 | 1 | 1 | 21 | 3 | 3 | 37 | 5 | 5 | 53 | 6 | 6 |
> | 6 | 1 | 1 | 22 | 3 | 3 | 38 | 5 | 5 | 54 | 6 | 6 |
> | 7 | 1 | 1 | 23 | 3 | 3 | 39 | 5 | 5 | 55 | 6 | 6 |
> | 8 | 1 | 1 | 24 | 4 | 4 | 40 | 5 | 5 | 56 | 6 | 6 |
> | 9 | 1 | 1 | 25 | 3 | 3 | 41 | 5 | 5 | 57 | 7 | 7 |
> | 10 | 1 | 1 | 26 | 4 | 4 | 42 | 6 | 6 | 58 | 6 | 6 |
> | 11 | 1 | 1 | 27 | 4 | 4 | 43 | 5 | 5 | 59 | 6 | 6 |
> | 12 | 1 | 1 | 28 | 4 | 4 | 44 | 5 | 5 | 60 | 6 | 6 |
> | 13 | 1 | 1 | 29 | 4 | 4 | 45 | 5 | 5 | 61 | 6 | 6 |
> | 14 | 2 | 2 | 30 | 4 | 4 | 46 | 5 | 5 | 62 | 7 | 7 |
> | 15 | 3 | 3 | 31 | 4 | 4 | 47 | 6 | 6 | 63 | 6 | 6 |
> | 16 | 2 | 2 | 32 | 4 | 4 | 48 | 6 | 6 | 64 | 6 | 6 |
>
> ## Memory overhead-latency trade-off (W2)
>
> We appreciate the reviewer’s comment and agree that memory overhead is an important factor when incorporating KV caching.
> MARché explores a latency–memory trade-off: it increases memory usage in exchange for faster inference, following a design philosophy similar to widely adopted KV caching in autoregressive models.
>
> In practice, the memory footprint introduced by MARché remains manageable.
> MAR models are relatively compact compared to large language models, and all cached data fits comfortably within the memory of a single modern GPU. Moreover, recent advances in KV cache compression and eviction techniques could be integrated in future work to further reduce memory consumption.
>
> For completeness, we report below the peak FP16 GPU memory usage of MAR and MARché across model scales.
>
> |Model Size|Model|Param|Peak Mem (GB)|
> |--|--|--|--|
> |Base|MAR-B|208M|27.58|
> ||MARché-B|208M|30.58|
> |Large|MAR-L|479M|32.07|
> ||MARché-L|479M|40.12|
> |Huge|MAR-H|943M|40.56|
> ||MARché-H|943M|53.98|

---

> ### Author Response · Authors · 2025-11-21
> **Response to Reviewer 54c7 (2/2)**
>
> ## Scalability of MARché (W3)
>
> We agree that demonstrating results on additional high-resolution datasets would further strengthen the case for the generalizability of our approach.
> However, our baseline model, MAR, is trained specifically on ImageNet 256×256. Evaluating at higher resolutions would therefore require training new MAR models from scratch at 512×512 or 1024×1024, which is currently beyond our computational budget.
>
> However, we actually expect MARché to yield even larger efficiency gains at higher resolutions.
> As image resolution increases, the number of tokens, and thus both the FLOPs and the total decoding cost, grow substantially.
> In such regimes, MARché can cache and reuse representations for a larger fraction of tokens across steps, which should translate into greater reductions in redundant computation.
>
> We appreciate the reviewer raising this important point, and we view a thorough evaluation of MARché on higher-resolution datasets as valuable future work.
>
> ## Comparison with LazyMAR (W4)
>
> We appreciate the reviewer’s suggestion and have conducted a comprehensive comparison against LazyMAR under the exact same experimental settings as our MARché evaluation.
> As shown in the table below, MARché consistently outperforms LazyMAR across all model scales.
> While LazyMAR achieves a 1.41×–1.53× speedup, MARché obtains a larger 1.57×–1.72× speedup, along with better image quality (lower FID and higher IS) in every configuration.
> These results further strengthen our claim that MARché provides a more favorable quality–latency trade-off than existing training-free MAR acceleration methods.
>
> Since the LazyMAR source code was released on October 24, after our submission deadline, it was not feasible to include a comparison at submission time.
> We will include these results in the revised version of the paper, along with a Pareto curve to more clearly highlight the quality–speed trade-off between LazyMAR and MARché.
>
> |Model Size|Model|Latency|FID ↓|IS ↑|Speedup|
> |--|--|--|--|--|--|
> |Base|LazyMAR-B|0.074|5.32|235.1|1.41×|
> ||MARché-B|0.064|2.56|270.3|1.57×|
> |Large|LazyMAR-L|0.132|4.32|246.6|1.46×|
> ||MARché-L|0.115|2.16|278.6|1.68×|
> |Huge|LazyMAR-H|0.220|4.00|251.9|1.53×|
> ||MARché-H|0.195|2.02|281.4|1.72×|
>
>
> ## Robustness of MARché Under Abrupt Content Transitions (W5, Q2)
>
> As the reviewer noted, certain stages of MAR decoding, especially transitions from background regions to semantically richer foreground structures, can exhibit larger changes in token representations. Our analysis in Figures 11 and 12 indeed shows that the spatial distribution of high-variation tokens differs between early and later decoding steps.
>
> However, two observations suggest that such abrupt content transitions do not significantly harm MARché’s efficiency or image quality.
>
> First, although the locations of high-variation tokens shift across steps, the number of such tokens remains consistently small across layers and decoding stages. This means that even when the model generates structurally different content, the proportion of tokens that require refresh remains within the capacity of our active set.
>
> Second, our method always refreshes the top-ranked unstable tokens at every step, regardless of where they occur in the image. Therefore, if a region undergoes a sudden semantic transition, those tokens naturally receive priority in the refresh set. In this sense, the selective-refresh mechanism dynamically adapts to abrupt content changes without requiring an explicit stage-dependent schedule.
>
> Together, these observations suggest that while abrupt content transitions are an important consideration, MARché is sufficiently robust to handle them without compromising efficiency or image quality.

---

### Official Review · Reviewer_5nNx · 2025-10-29

**Soundness:** 3
**Presentation:** 3
**Contribution:** 2
**Rating:** 4
**Confidence:** 4

**Summary:**

The paper introduces MARché, a training-free decoding scheme that speeds up masked autoregressive (MAR) image generation by avoiding redundant recomputation. It splits tokens each step into an active set (newly generated + a few contextually relevant “refresh” tokens) and a cached set whose key/value projections are reused via cache-aware attention; relevance is decided by attention from newly generated tokens (selective KV refresh). This preserves full-context modeling (using an online-softmax merge that’s equivalent to standard attention) while skipping unnecessary FFN work. On ImageNet 256×256, MARché reports up to ~1.7× latency speedup vs. MAR with only minor FID/IS changes, and requires no architecture changes or retraining.

**Strengths:**

Training-Free Speedup: MARché offers a significant speedup in masked autoregressive (MAR) image generation without requiring any retraining, making it highly efficient for real-time applications.

Efficient Use of Cache: By reusing key/value projections from previous steps through cache-aware attention, the method avoids redundant calculations, which reduces computational load while preserving high-quality generation.

No Architecture Modifications: It doesn't require any changes to the underlying model architecture, making it easy to integrate into existing systems.

Performance Gains: The paper reports a 1.7× latency reduction with only minor decreases in image quality metrics like FID (Fréchet Inception Distance) and IS (Inception Score), showing that it delivers efficiency without significantly compromising performance.

Broad Applicability: The approach can be applied to other autoregressive models without much modification, offering potential for wide adoption in various image generation tasks.

**Weaknesses:**

1 The acceleration gains from the proposed MARché method are modest. Despite achieving a 1.7× speedup, the improvement may not be significant enough for certain applications.

2 The method was primarily tested on the ImageNet 256×256 dataset. Its performance and acceleration effect on larger-scale tasks, such as high-resolution images, video generation, or multi-modal data, remain unclear. If the method does not perform well in these scenarios, its general applicability could be limited.

3 The method lacks significant novelty, as it primarily builds on existing techniques like cache management and selective KV refresh, and its scope is limited to optimizing a specific approach in the rapidly advancing field of autoregressive image generation.

**Questions:**

see above

---

> ### Author Response · Authors · 2025-11-21
> **Response to Reviewer 5nNx (1/2)**
>
> ## Magnitude of Speedup (W1)
>
> We understand the reviewer’s concern that a 1.7× acceleration may appear modest when viewed in isolation. However, the achievable speedup must be interpreted in the context of bidirectional masked autoregressive transformers, where acceleration is fundamentally constrained by the model architecture. In MAR, each decoding step requires full bidirectional attention over an ever-growing set of visible tokens, leaving limited room for eliminating computation.
>
> Within these constraints, MARché removes a substantial portion of redundant attention computation while preserving image quality. This results in consistent 1.57×–1.72× real-world latency gains across all model sizes. Additionally, MARché outperforms the most recent training-free accelerator, LazyMAR, both in latency and image quality, confirming that these gains are meaningful in practice.
> The table below summarizes the comparison between LazyMAR and MARché.
>
> |Model Size|Model|Latency|FID ↓|IS ↑|Speedup|
> |--|--|--|--|--|--|
> |Base|LazyMAR-B|0.074|5.32|235.1|1.41×|
> ||MARché-B|0.064|2.56|270.3|1.57×|
> |Large|LazyMAR-L|0.132|4.32|246.6|1.46×|
> ||MARché-L|0.115|2.16|278.6|1.68×|
> |Huge|LazyMAR-H|0.220|4.00|251.9|1.53×|
> ||MARché-H|0.195|2.02|281.4|1.72×|
>
> ## Scalability of MARché (W2)
>
> We agree that evaluating MARché on higher-resolution or more complex tasks would further substantiate its effectiveness. However, the largest publicly available pretrained MAR models operate at 256×256 resolution. Extending the study to higher resolutions would require retraining MAR itself, which is currently beyond our hardware capacity.
>
> Even so, we expect MARché’s benefits to become more pronounced at higher resolutions. As resolution increases, the number of tokens grows substantially, and the cost of repeatedly recomputing attention rises accordingly. In such scenarios, MARché can reuse a larger portion of tokens across steps, reducing redundant computation and likely yielding even greater acceleration than what we observe at 256×256.
>
> We consider exploring higher-resolution and multi-modal settings an important direction for future work, and we thank the reviewer for highlighting this point.

---

> ### Author Response · Authors · 2025-11-21
> **Response to Reviewer 5nNx (2/2)**
>
> ## Novelty and Architectural Scope (W3)
>
> **Novelty:**
> We appreciate the reviewer’s comment and would like to clarify the novelty of our contribution. While MARché draws inspiration from ideas such as KV caching and selective refresh, its key novelty lies in making these techniques applicable and effective within masked generative modeling—something fundamentally non-trivial.
>
> The core difficulty is that masked generative models rely on full bidirectional attention, where every token interacts with all others at each decoding step. As a result, the key and value representations of generated tokens evolve over time, making naive caching ineffective or even harmful. This stands in sharp contrast to causal autoregressive language models, where KV caching is straightforward due to the strictly growing context.
>
> MARché resolves this challenge by leveraging the architectural properties of MAR and introducing mechanisms that allow KV caching to function reliably in this setting. Our analysis shows that although KV representations do fluctuate across steps, these fluctuations are sparse, enabling safe reuse when paired with selective refresh.
>
> To operationalize this, MARché introduces (1) a principled separation of tokens into active and cached sets based on MAR’s fixed generation order, and (2) cache-aware attention that decouples their computation paths and reduces unnecessary memory and compute overhead. This design enables KV caching to be integrated naturally into masked autoregressive generation, yielding substantial efficiency gains while preserving image quality.
>
> **Scope:**
> We would like to emphasize that existing masked or iterative generative models, such as MaskGIT, were not designed with KV caching in mind. In such models, all previously generated tokens are recomputed at every iteration, and their hidden representations vary significantly across cycles, fundamentally preventing KV reuse. Thus, KV-cache techniques developed for causal autoregressive transformers cannot be directly applied to bidirectional masked autoregressive settings.
>
> In contrast, MAR exhibits structural properties that create a previously unexplored opportunity for KV reuse: a fixed generation order and a monotonically expanding set of generated tokens. MARché exploits these properties to enable KV caching even under full bidirectional attention, as described above. We agree that MARché is directly applicable only to architectures that share these characteristics—namely, a fixed generation order and reasonably stable generated tokens.
>
> However, we believe that enabling efficient KV caching in the current state-of-the-art masked autoregressive image generator is itself a meaningful contribution, given that bidirectional masked models have long been assumed incompatible with caching. Moreover, the insights behind MARché, exploiting structured stability in token representations, suggest architectural principles that could guide the design of future iterative masked generative models to support KV caching. Exploring such redesigns represents a promising direction for future work.

---

> > ### Comment · Reviewer_5nNx · 2025-11-22
> >
> > Thank you for your response. It addresses most of my concerns. Although I understand that better results can be achieved at a larger scale, within the scope of AR image generation, choosing different pipelines (RAR, LPD) may be much more effective than optimizing a single pipeline alone. I will first raise the score to 6 and then see what the other reviewers say.

---

### Official Review · Reviewer_TpJn · 2025-11-02

**Soundness:** 2
**Presentation:** 3
**Contribution:** 2
**Rating:** 4
**Confidence:** 4

**Summary:**

This paper proposes MARCHÉ, a training-free acceleration method for masked autoregressive (MAR) image generation. The key idea is to reuse KV representations that remain stable across decoding steps by maintaining cached tokens and selectively refreshing active ones via attention signals and online softmax. The method reports up to ~1.7× speedup on MAR models with minor quality changes and requires no model modification.

**Strengths:**

- Training-free & plug-and-play: No architectural change, no retraining, practical for deployment.
- Clear and practical idea: Efficient KV reuse tailored to masked autoregressive generation.

**Weaknesses:**

1. Lack of evaluation across different MAR schedules
  - The KV-reuse assumption is tested under a single MAR step schedule.
  - It is unclear whether the method generalizes to different decoding steps.

2. KV-cache strategy generality remains uncertain
  - Only evaluated on MAR.
  - It is unknown how well the approach applies to other iterative masked Autoregressive generative frameworks.

3. Comparison to LazyMAR needs clarification
  - LazyMAR is also a training-free MAR acceleration method. However, the paper does not clearly demonstrate superior performance over LazyMAR (e.g., matched-speed or matched-quality comparisons). A direct quality-speed Pareto comparison would be necessary to substantiate claims of advantage.

**Questions:**

1. KV Memory Layout Question
Have you explored pre-allocating a single contiguous KV buffer and directly writing both cached tokens and newly active tokens into it during loading, instead of handling them in separate buffers and fusing the output? In principle, constructing a contiguous KV block upfront could eliminate the concatenate step and enable one attention kernel call on a regularized sequence length, which may yield higher efficiency.

2. Question on the “generation order” claim
I am not a domain expert, but from reading LazyMAR, my understanding is that it keeps the predefined MAR decoding schedule (i.e., which tokens are decoded at step t) and only dynamically decides which tokens can reuse features within that step to avoid recomputation. In other words, LazyMAR seems to reuse features across layers but does not change which tokens are revealed at each decoding step.
Given this, I am not fully understanding the statement that LazyMAR “departs from MAR’s predefined generation order.” Could you clarify in what sense the generation order is altered? If LazyMAR truly modifies the token decoding order, it would be helpful to reference the specific mechanism or evidence.

---

> ### Author Response · Authors · 2025-11-21
> **Response to Reviewer TpJn (1/2)**
>
> ## Evaluation across different MAR schedules (W1)
>
> To assess the generality of MARché beyond the default MAR schedule, we additionally evaluated it under a 96-step decoding schedule, which uses a different unmasking trajectory.
> Across all model scales (Base, Large, Huge), MARché continues to provide a consistent latency–quality trade-off: it achieves 1.6–1.8× speedup while maintaining competitive FID and IS, showing the same pattern as observed under the original MAR schedule.
>
> These results suggest that MARché is not tied to a specific step schedule and generalizes well to alternative MAR decoding dynamics. We will include these results and a detailed discussion in the revised version of the paper.
>
> |Model Size|Model|Latency|FID ↓|IS ↑|Speedup|
> |--|--|--|--|--|--|
> |Base|MAR-B|0.104|2.34|280.5||
> ||MARché-B|0.093|2.49|269.3|1.66×|
> |Large|MAR-L|0.286|1.81|303.4||
> ||MARché-L|0.157|2.15|276.2|1.82×|
> |Huge|MAR-H|0.505|1.59|300.8||
> ||MARché-H|0.313|1.74|293.5|1.61×|
>
>
> ## General applicability of MARché (W2)
>
> Thank you for raising this important question regarding the generality of our KV-cache strategy.
>
> First, we would like to clarify that existing iterative masked generative models, such as MaskGIT, were not designed with KV caching in mind.
> In these models, previously generated tokens are fully recomputed in every iteration, and their hidden representations change substantially across cycles, making KV reuse infeasible.
>
> In contrast, MAR—although also not originally designed for caching—exhibits structural properties that create an opportunity for KV reuse, namely a fixed generation order and a monotonically expanding set of generated tokens.
> Once a token is unmasked in MAR, it remains unmasked for the rest of the decoding process, providing a stable generation trajectory.
> However, due to MAR’s bidirectional attention, the KV representations of previously generated tokens still fluctuate as new tokens are introduced.
> Our analysis (Figures 11 and 12) shows that these fluctuations are sparse, which is precisely the observation that MARché builds upon.
>
> The contribution of MARché lies in addressing this challenge: we introduce a principled mechanism to (1) identify which tokens truly require recomputation and which can be safely cached, and (2) perform efficient attention by separating tokens into cached and active sets.
> This allows KV caching to function reliably even under full bidirectional attention, which is typically considered incompatible with caching.
>
> Since MAR represents the current state-of-the-art masked autoregressive framework for image generation, we believe that enabling efficient KV caching within this setting is itself a meaningful contribution.
> Our method demonstrates that bidirectional masked generative models, which have traditionally been viewed as incompatible with caching, can, in fact, benefit significantly from it when analyzed appropriately.
>
> Finally, we agree that MARché is directly applicable only to MAR-like architectures—those with a fixed generation order and reasonably stable token representations.
> However, we believe that other iterative masked generative models could support KV caching if their architectures were modified to encourage token stability or adopt a predictable generation schedule.
> Exploring such redesigns represents a promising direction for future work.
>
> ## Comparison to LazyMAR (W3)
>
> We appreciate the reviewer’s suggestion and have conducted a comprehensive comparison against LazyMAR under the exact same experimental settings as our MARché evaluation.
> As shown in the table below, MARché consistently outperforms LazyMAR across all model scales.
> LazyMAR achieves a 1.41×–1.53× speedup, whereas MARché achieves a larger 1.57×–1.72× speedup while also delivering better image quality (lower FID and higher IS) in all configurations.
> These results further support our claim that MARché provides a more favorable quality–latency trade-off compared to existing training-free MAR acceleration methods.
>
> Since the LazyMAR source code was released on October 24, after our submission deadline, it was not feasible to include a comparison at submission time.
> We will include these results in the revised version of the paper, along with a Pareto curve to more clearly highlight the quality–speed trade-off between LazyMAR and MARché.
>
> |Model Size|Model|Latency|FID ↓|IS ↑|Speedup|
> |--|--|--|--|--|--|
> |Base|LazyMAR-B|0.074|5.32|235.1|1.41×|
> ||MARché-B|0.064|2.56|270.3|1.57×|
> |Large|LazyMAR-L|0.132|4.32|246.6|1.46×|
> ||MARché-L|0.115|2.16|278.6|1.68×|
> |Huge|LazyMAR-H|0.220|4.00|251.9|1.53×|
> ||MARché-H|0.195|2.02|281.4|1.72×|

---

> ### Author Response · Authors · 2025-11-21
> **Response to Reviewer TpJn (2/2)**
>
> ## KV Memory Layout Implementation (Q1)
>
> Thank you for the insightful suggestion regarding KV memory layout.
> Pre-allocating a single contiguous KV buffer is indeed a possible implementation. However, in our setting it introduces an additional indexing overhead that we aim to avoid.
>
> Our cache-aware attention uses two separate computation paths, one for active tokens and one for cached tokens, each of which requires retrieving a different subset of KV entries. If both sets are stored in a single contiguous buffer, each access would require per-token indexing to filter out the relevant positions. This additional indirection occurs every time the attention kernel loads KV values, potentially offsetting the expected efficiency gain.
>
> To minimize such overhead, we instead maintain two dedicated buffers and perform token selection once before computation. This design requires only a single indexing step during the separation phase, after which both attention paths operate on clean, contiguous KV segments without further indirection.
>
> ## Generation order and LazyMAR (Q2)
>
> We agree that LazyMAR does not explicitly define a new decoding schedule. Our point is that its dynamic token-selection mechanism can implicitly diverge from MAR’s predetermined schedule.
>
> In MAR, the generation order is fixed in advance, and each step specifies a particular set of tokens that must be generated. LazyMAR, however, selects which tokens to update at each step based solely on cosine-similarity scores, independent of MAR’s schedule. Consequently, the tokens updated at step t may not align with the tokens MAR intends for that step, meaning that some tokens may be generated using stale contextual information cached from previous steps.
> This implicit deviation from the prescribed generation order may contribute to quality degradation.
>
> Thus, our statement refers to this mismatch: although LazyMAR preserves MAR’s overall structure, its dynamic selection policy can cause tokens to be updated at different steps than those specified by MAR’s original schedule.

---

### Meta-Review · Area_Chair_uhg4 · 2026-01-07

**Summary:**

Most initial scores were negative or on the fence. The main issues were limited generality beyond MAR and no validation on high resolution data. TpJn and 5nNx both said the fixed MAR generation order makes the approach hard to use outside this setup. 54c7 also noted the method depends on hand tuned hyperparameters, with no clear rule for choosing them. In rebuttal, the authors did not add results beyond 256×256, and they did not give a principled way to make the hyperparameters less brittle. I suggest addressing the scalability and generality gaps and then resubmitting to another venue.

**Reviewer Concerns:**

Addressed:

LazyMAR: They added a comparison and show a better speed quality tradeoff, with a 1.57× to 1.72× speedup over LazyMAR.

Memory: They reported peak memory going from 40.56GB to 53.98GB on the Huge model.

Details: They clarified the memory layout and how active vs cached tokens are handled for TpJn.

Outstanding:

High resolution: Results are only on ImageNet 256×256. They said they cannot retrain at 512×512 or 1024×1024 due to hardware limits, so scaling is still untested.

Generality: The approach is still tied to MAR and its fixed order.

Hyperparams: Key settings are still chosen by trial and error, like using layer 2 for scoring and refreshing every 3 steps. There is no clear rule for choosing them, and it is not clear they will hold across data.

**Reviewer Scores:**

TpJn: Likely 4. The LazyMAR comparison helps, but the method still does not generalize beyond MAR style decoding.

5nNx: Likely 6 after rebuttal.

54c7: Likely 4. Hyperparameter choices are still hand tuned and there are no high resolution experiments.

L52V: Likely 6. They like the speedup, but the dependence on MAR is still a real weakness.

---

### Decision · Program_Chairs · 2026-01-26

Reject